# Enhancing Kernel Flexibility via Learning Asymmetric Locally-Adaptive Kernels

## Abstract

The lack of sufficient flexibility is the key bottleneck of kernel-based learning that relies on manually designed, pre-given, and non-trainable kernels. To enhance kernel flexibility, this paper introduces the concept of Locally-Adaptive-Bandwidths (LAB) as trainable parameters to enhance the Radial Basis Function (RBF) kernel, giving rise to the LAB RBF kernel. The parameters in LAB RBF kernels are data-dependent, and its number can increase with the dataset, allowing for better adaptation to diverse data patterns and enhancing the flexibility of the learned function. This newfound flexibility also brings challenges, particularly with regards to asymmetry and the need for an efficient learning algorithm. To address these challenges, this paper for the first time establishes an asymmetric kernel ridge regression framework and introduces an iterative kernel learning algorithm. This novel approach not only reduces the demand for extensive support data but also significantly improves generalization by training bandwidths on the available training data. Experimental results on real datasets underscore the outstanding performance of the proposed algorithm, highlighting its superior capability in reducing the required number of support data compared to Nyström approximation-based algorithms, all while maintaining state-of-the-art regression accuracy.

## 1 Introduction

Kernel methods play a foundational role within the machine learning community, offering a lot of classical non-linear algorithms, including Kernel Ridge Regression (KRR, Vovk (2013)), Support Vector Machines (SVM, Cortes & Vapnik (1995)), kernel principal component analysis (Schölkopf et al., 1997), and a host of other innovative algorithms. Nowadays, kernel methods maintain their importance thanks to their interpretability, strong theoretical foundations, and versatility in handling diverse data types (Ghorbani et al., 2020; Bach, 2022; Jerbi et al., 2023). However, as newer techniques like deep learning gain prominence, kernel methods reveal a shortcoming: the learned function's flexibility often falls short of expectations.

Recent studies (Ma et al., 2017; Montanari & Zhong, 2020) demonstrate that the fundamental behavior of a sufficiently flexible model, such as deep models, can interpolate samples while maintaining good generalization ability. However, interpolation with either single or multi-kernel methods is achieved using kernels close to Dirac function and ridgeless models, leading to large parameter norms and poor generalization. The reason for imperfect interpolation in kernel-based learning stems from its inherent lack of flexibility. The flexibility of a model, often referred to as its degree of freedom, is directly indicated by the number of its free parameters. Recent studies (Allen-Zhu et al., 2019; Zhou & Huo, 2024) have revealed that a model with ample flexibility tends to be over-parameterized. However, the number of free parameters of classical kernel-based models are constrained by the number of training data points $N$, falling far short of the capabilities observed in over-parameterized deep models. The primary challenge in augmenting the trainable parameters of kernel methods arises from the reliance on manually designed, fixed kernels in traditional learning algorithms, which are inherently untrainable.

Over the past decade, various data-driven approaches have been explored to introduce trainable parameters to kernels to enhance the flexibility. However, despite significant progress, there remains ample room for improvement in this area. For instance, multiple kernel learning (MKL, Gönen & Alpaydın (2011)) employs linear combinations of kernels instead of a single one, yet the increase in

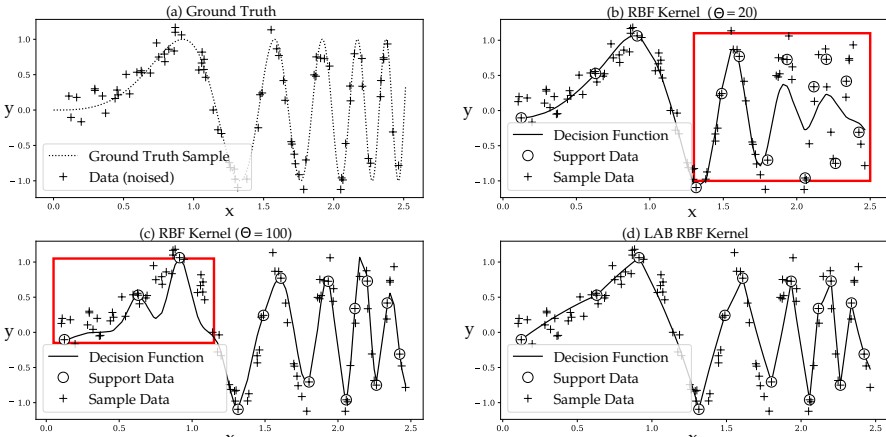

Figure 1: A toy example demonstrating the regression of a one-dimensional signal $y = \sin(2x^3)$. (a) Ground truth. (b) Function obtained with a universal kernel bandwidth, showing a lack of accuracy in high-frequency regions. (c) Function obtained with a increased bandwidth, resulting in unnecessary sharp changes in low-frequency areas. (d) In our proposed LAB RBF kernels, data-dependent bandwidths are trained, effectively adapting to the underlying function: larger bandwidths correspond to the left part, and smaller bandwidths correspond to the right part.

the number of learnable parameters is limited to the number of kernel candidates, falling short of expectations. Deep kernel learning (Wilson et al. (2016)) introduces deep architectures as explicit feature mappings for kernels, with the core of its techniques residing in feature learning rather than kernel learning. Recent works, such as Liu et al. (2020), propose the direct introduction of trainable parameters into the kernel matrix but lack corresponding kernel function formulations. Furthermore, the increased flexibility in these works, while advantageous, has not mitigated the computational issues arising from the large $N \times N$ kernel matrices.

In this paper, we introduce a novel approach to enhance kernel flexibility, significantly differing from current research endeavors. Our proposal involves augmenting the RBF kernel by introducing locally adaptive bandwidths. Given a dataset $\mathcal{X} = \{\boldsymbol{x}_1, \cdots, \boldsymbol{x}_N\} \subset \mathcal{R}^M$ and let $\odot$ denote the Hadamard product, then the kernel defined on $\mathcal{X}$ is outlined below:

$$\mathcal{K}(\boldsymbol{t}, \boldsymbol{x}_i) = \exp\left\{-\|\boldsymbol{\theta}_i \odot (\boldsymbol{t} - \boldsymbol{x}_i)\|_2^2\right\}, \qquad \forall \boldsymbol{x}_i \in \mathcal{X}, \ \forall \boldsymbol{t} \in \mathcal{R}^M, \tag{1}$$

where $\boldsymbol{\theta}_i \in \mathcal{R}_+^M, \forall i$ denotes a positive bandwidth[1]. We name (1) as Local-Adaptive-Bandwidth RBF (LAB RBF) kernels. The key difference between LAB RBF kernels and conventional RBF kernels lies in assigning distinct bandwidths $\boldsymbol{\theta}_i$ to each sample $\boldsymbol{x}_i$ rather than using a uniform bandwidth across all data points, and we proposed to estimate these bandwidths $\boldsymbol{\theta}_i$ from training data.

Obviously, the introduction of such data-dependent bandwidths can effectively increase the flexibility of kernel-based learning, making bandwidths adaptive to data, not universal as in conventional kernels. Fig. 1 illustrates the benefits of introducing locally-adaptive bandwidths. The underlying function $y = \sin(2x^3)$ exhibits varying frequencies across its domain. When employing the RBF kernel with a global bandwidth, a dilemma arises: using a small bandwidth (=20) yields an inadequate approximation of the high-frequency portion (highlighted in the red box in Fig.1 (b)). Conversely, a larger bandwidth (=100) is necessary to approximate the high-frequency section, resulting in a final function that is overly sharp and struggles to accurately represent the smooth portions (highlighted in the red box in Fig.1 (c)). The proposed LAB RBF kernel emerges as an optimal solution, offering a more flexible approach to bandwidths. As depicted in Fig. 1 (d), it strategically employs larger bandwidths on the left and smaller bandwidths on the right.

---

[1]Strictly speaking, the bandwidths in a LAB RBF kernel should be a vector function defined on $\mathcal{R}^M$. However, for better clarity in illustrating the subsequent learning algorithm, we discretely define the bandwidth for each support vector data point in a point by point way.

Notably, LAB RBF kernels exhibit inherent asymmetry since $\boldsymbol{\theta}_i$ is not required to be equal to $\boldsymbol{\theta}_j$, allowing for cases where $\mathcal{K}(\boldsymbol{x}_i, \boldsymbol{x}_j) \neq \mathcal{K}(\boldsymbol{x}_j, \boldsymbol{x}_i)$. This asymmetry, while providing added flexibility to LAB RBF kernels compared to traditional symmetric kernels, presents two algorithmic challenges: *how to determine the bandwidths for support data?* and *how to incorporate asymmetric LAB RBF kernels into existing kernel-based models?* This paper addresses these challenges by developing an asymmetric KRR model and introducing an innovative technique for learning asymmetric kernels directly from the training dataset. To summarize, the contributions of this paper are as follows:

**Flexible and trainable kernels.** The introduced LAB RBF kernel, as presented in (1), incorporates individualized bandwidths for each data point, introducing asymmetry and thereby augmenting the number of trainable parameters. This augmentation significantly boosts the model's flexibility when employing LAB RBF kernels in kernel-based learning, enabling it to better accommodate a wide array of diverse data patterns.

**Asymmetric kernel ridge regression framework.** For the application of asymmetric LAB RBF kernels, this paper for the first time establishes an asymmetric KRR framework. An analytical expression for the stationary points is derived, elegantly represented as a linear combination of function evaluations at training data. Remarkably, coefficients of the combination take the same form as those of classical symmetric KRR models, despite the asymmetric nature of the kernel matrix.

**Robust kernel learning algorithm.** We introduce a novel kernel learning algorithm tailored for LAB RBF kernels, enabling the determination of local bandwidths. This algorithm empowers the regression function not only to interpolate support data effectively but also to achieve excellent generalization ability by tuning bandwidths on the training data.

The experimental results underscore the advanced performance of our algorithm: it achieves state-of-the-art regression accuracy while substantially reducing the required amount of support data (model complexity) compared to advanced kernel-based methods.

## 2 ASYMMETRIC KERNEL RIDGE REGRESSION

### 2.1 KERNEL RIDGE REGRESSION

Kernel ridge regression (Vovk, 2013) is one of the most elementary kernelized algorithms. Define the dataset $\mathcal{X} = \{\boldsymbol{x}_1, \cdots, \boldsymbol{x}_N\} \subset \mathcal{R}^M, \mathcal{Y} = \{y_1, \cdots, y_N\} \subset \mathcal{R}$, and data matrix $\boldsymbol{X} = [\boldsymbol{x}_1, \boldsymbol{x}_2, \cdots, \boldsymbol{x}_N] \in \mathcal{R}^{M \times N}, \boldsymbol{Y} = [y_1, y_2, \cdots, y_N]^\top \in \mathcal{R}^N$. The task is to find a linear function in a high dimensional feature space, denoted as $\mathcal{R}^F$, which models the dependencies between the features $\phi(\boldsymbol{x}_i), \forall \boldsymbol{x}_i \in \mathcal{X}$ of input and response variables $y_i, \forall y_i \in \mathcal{Y}$. Here, $\phi : \mathcal{R}^M \to \mathcal{R}^F$ denotes the feature mapping from the data space to the feature space. Define $\phi(\boldsymbol{X}) = [\phi(\boldsymbol{x}_1), \phi(\boldsymbol{x}_2), \cdots, \phi(\boldsymbol{x}_N)]$, then the classical optimization model is as follow:

$$\min_{\boldsymbol{w}} \quad \frac{\lambda}{2} \boldsymbol{w}^\top \boldsymbol{w} + \frac{1}{2} \|\boldsymbol{Y} - \phi(\boldsymbol{X})^\top \boldsymbol{w}\|_2^2, \tag{2}$$

where $\lambda > 0$ is a trade-off hyper-parameter. By utilizing the following well-known matrix inversion lemma (see Petersen & Pedersen (2008); Murphy (2012) for more information),

$$(\boldsymbol{A} + \boldsymbol{B}\boldsymbol{D}^{-1}\boldsymbol{C})^{-1}\boldsymbol{B}\boldsymbol{D}^{-1} = \boldsymbol{A}^{-1}\boldsymbol{B}(\boldsymbol{C}\boldsymbol{A}^{-1}\boldsymbol{B} + \boldsymbol{D})^{-1}, \tag{3}$$

one can obtain the solution of KRR as follow

$$\boldsymbol{w}^* = (\phi(\boldsymbol{X})\phi(\boldsymbol{X})^\top + \lambda\boldsymbol{I}_F)^{-1}\phi(\boldsymbol{X})\boldsymbol{Y} \overset{(a)}{=} \phi(\boldsymbol{X})(\lambda\boldsymbol{I}_N + \phi(\boldsymbol{X})^\top\phi(\boldsymbol{X}))^{-1}\boldsymbol{Y},$$

where (3) is applied in (a) with $\boldsymbol{A} = \boldsymbol{I}_F, \boldsymbol{B} = \phi(\boldsymbol{X}), \boldsymbol{C} = \phi^\top(\boldsymbol{X}), \boldsymbol{D} = \boldsymbol{I}_N$.

### 2.2 ASYMMETRIC KERNEL RIDGE REGRESSION

Assume we have two feature mappings from data space to an unknown vector space: $\phi : \mathcal{R}^M \to \mathcal{R}^F$, and $\psi : \mathcal{R}^M \to \mathcal{R}^F$. Given training dataset $(\boldsymbol{X}, \boldsymbol{Y})$, the asymmetric kernel ridge regression model is

$$\min_{\boldsymbol{w}, \boldsymbol{v}} \lambda\boldsymbol{w}^\top\boldsymbol{v} + (\phi^\top(\boldsymbol{X})\boldsymbol{w} - \boldsymbol{Y})^\top(\psi^\top(\boldsymbol{X})\boldsymbol{v} - \boldsymbol{Y})$$

$$\iff \min_{\boldsymbol{w}, \boldsymbol{v}} \lambda\boldsymbol{w}^\top\boldsymbol{v} + \frac{1}{2}\|\phi^\top(\boldsymbol{X})\boldsymbol{w} - \boldsymbol{Y}\|_2^2 + \frac{1}{2}\|\psi^\top(\boldsymbol{X})\boldsymbol{v} - \boldsymbol{Y}\|_2^2 - \frac{1}{2}\|\psi^\top(\boldsymbol{X})\boldsymbol{v} - \phi^\top(\boldsymbol{X})\boldsymbol{w}\|_2^2.$$
$$\tag{4}$$

Here, $\lambda > 0$ serves as a trade-off hyper-parameter between the regularization term $\boldsymbol{w}^\top \boldsymbol{v}$ and the error term $(\phi^\top(\boldsymbol{X})\boldsymbol{w} - \boldsymbol{Y})^\top(\psi^\top(\boldsymbol{X})\boldsymbol{v} - \boldsymbol{Y})$. Given the existence of two feature mappings, we have two regressors in the space $\mathcal{R}^F$: $f_1(\boldsymbol{t}) = \phi^\top(\boldsymbol{t})\boldsymbol{w}$ and $f_2(\boldsymbol{t}) = \psi^\top(\boldsymbol{t})\boldsymbol{v}$. To enhance clarity regarding the meaning of the error term, we decompose it into the sum of three terms, as shown in the second line. The terms $\frac{1}{2}\|\phi^\top(\boldsymbol{X})\boldsymbol{w} - \boldsymbol{Y}\|_2^2 + \frac{1}{2}\|\psi^\top(\boldsymbol{X})\boldsymbol{v} - \boldsymbol{Y}\|_2^2$ are employed to minimize the regression error. Additionally, the term $\lambda\boldsymbol{w}^\top \boldsymbol{v} - \frac{1}{2}\|\psi^\top(\boldsymbol{X})\boldsymbol{v} - \phi^\top(\boldsymbol{X})\boldsymbol{w}\|_2^2$ aims to emphasize the substantial distinction between the two regressors. Then we have the following result on the stationary points.

**Theorem 1.** *One of the stationary points of (4) is*

$$\boldsymbol{w}^* = \psi(\boldsymbol{X})(\phi^\top(\boldsymbol{X})\psi(\boldsymbol{X}) + \lambda\boldsymbol{I}_N)^{-1}\boldsymbol{Y}, \qquad \boldsymbol{v}^* = \phi(\boldsymbol{X})(\psi^\top(\boldsymbol{X})\phi(\boldsymbol{X}) + \lambda\boldsymbol{I}_N)^{-1}\boldsymbol{Y}. \quad (5)$$

The proof is presented in Appendix A. Theorem 1 establishes a crucial result, demonstrating that the stationary points can still be represented as a linear combination of function evaluations on the training dataset. This validates the practical feasibility of the proposed framework. With the conclusion in Theorem 1, we can easily apply asymmetric kernel functions. Define an asymmetric kernel by the inner product of $\phi$ and $\psi$, i.e. $\mathcal{K}(\boldsymbol{x}, \boldsymbol{t}) = \langle \phi(\boldsymbol{x}), \psi(\boldsymbol{t}) \rangle$, $\forall \boldsymbol{x}, \boldsymbol{t} \in \mathcal{R}^M$, and denote a kernel matrix $[\boldsymbol{K}(\boldsymbol{X}, \boldsymbol{X})]_{ij} = \mathcal{K}(\boldsymbol{x}_i, \boldsymbol{x}_j) = \phi^\top(\boldsymbol{x}_i)\psi(\boldsymbol{x}_j)$, $\forall \boldsymbol{x}_i, \boldsymbol{x}_j \in \mathcal{X}$, then we obtain two regression functions:

$$\begin{aligned}
f_1(\boldsymbol{t}) &= \phi(\boldsymbol{t})^\top \boldsymbol{w}^* = \boldsymbol{K}(\boldsymbol{t}, \boldsymbol{X})(\boldsymbol{K}(\boldsymbol{X}, \boldsymbol{X}) + \lambda\boldsymbol{I}_N)^{-1}\boldsymbol{Y}, \\
f_2(\boldsymbol{t}) &= \psi(\boldsymbol{t})^\top \boldsymbol{v}^* = \boldsymbol{K}^\top(\boldsymbol{X}, \boldsymbol{t})(\boldsymbol{K}^\top(\boldsymbol{X}, \boldsymbol{X}) + \lambda\boldsymbol{I}_N)^{-1}\boldsymbol{Y}.
\end{aligned} \quad (6)$$

Theorem 1 also indicates the proposed asymmetric KRR framework includes the symmetric one. That is, model (4) and model (2) share the same stationary points when the two feature mappings are equivalent, as shown in the following corollary.

**Corollary 2.** *If the two feature mappings $\phi$ and $\psi$ are equivalent, i.e. $\phi(\boldsymbol{x}) = \psi(\boldsymbol{x}), \forall \boldsymbol{x} \in \mathcal{R}^M$, then stationary conditions of the asymmetric KRR model (4) and the symmetric KRR model (2) are equivalent. And the stationary point is $\boldsymbol{w}^* = \boldsymbol{v}^* = \phi(\boldsymbol{X})(\lambda\boldsymbol{I}_N + \phi(\boldsymbol{X})^\top\phi(\boldsymbol{X}))^{-1}\boldsymbol{Y}$.*

## 2.3 ALTERNATIVE DERIVATION AND FUNCTION EXPLANATION

We can also derive a similar result in Theorem 1 in a LS-SVM-like approach (Suykens & Vandewalle, 1999), from which we can better understand the relationship between the two regression functions. By introducing error variables $e_i = y_i - \phi(\boldsymbol{x}_i)^\top \boldsymbol{w}$ and $r_i = y_i - \psi(\boldsymbol{x}_i)^\top \boldsymbol{v}$, the last term in (4) equals to $\sum_i e_i r_i$. According to this result, we have the following optimization:

$$\begin{aligned}
\min_{\boldsymbol{w}, \boldsymbol{v}, \boldsymbol{e}, \boldsymbol{r}} \quad & \lambda\boldsymbol{w}^\top \boldsymbol{v} + \sum_{i=1}^{N} e_i r_i \\
\text{s.t.} \quad & e_i = y_i - \phi(\boldsymbol{x}_i)^\top \boldsymbol{w}, \quad \forall i = 1, 2, \cdots, N, \\
& r_i = y_i - \psi(\boldsymbol{x}_i)^\top \boldsymbol{v}, \quad \forall i = 1, 2, \cdots, N.
\end{aligned} \quad (7)$$

From the Karush-Kuhn-Tucker (KKT) conditions (Boyd & Vandenberghe, 2004), we can obtain the following result on the KKT points.

**Theorem 3.** *Let $\boldsymbol{\alpha} = [\alpha_1, \cdots, \alpha_N]^\top \in \mathcal{R}^N$ and $\boldsymbol{\beta} = [\beta_1, \cdots, \beta_N]^\top \in \mathcal{R}^N$ be Lagrange multipliers of constraints $e_i = y_i - \phi(\boldsymbol{x}_i)^\top \boldsymbol{w}$ and $r_i = y_i - \psi(\boldsymbol{x}_i)^\top \boldsymbol{v}, \forall i = 1, \cdots, N$, respectively. Then one of the KKT points of (7) is*

$$\begin{aligned}
\boldsymbol{w}^* &= \frac{1}{\lambda}\psi(\boldsymbol{X})\boldsymbol{\beta}^*, & \boldsymbol{e}^* &= \boldsymbol{\beta}^* = \lambda(\phi^\top(\boldsymbol{X})\psi(\boldsymbol{X}) + \lambda\boldsymbol{I}_N)^{-1}\boldsymbol{Y}, \\
\boldsymbol{v}^* &= \frac{1}{\lambda}\phi(\boldsymbol{X})\boldsymbol{\alpha}^*, & \boldsymbol{r}^* &= \boldsymbol{\alpha}^* = \lambda(\psi^\top(\boldsymbol{X})\phi(\boldsymbol{X}) + \lambda\boldsymbol{I}_N)^{-1}\boldsymbol{Y}.
\end{aligned}$$

This model shares a close relationship with existing models. For instance, by modifying the regularization term from $\boldsymbol{w}^\top \boldsymbol{v}$ to $\boldsymbol{w}^\top \boldsymbol{w} + \boldsymbol{v}^\top \boldsymbol{v}$ and flipping the sign of $\sum_{i=1}^{N} e_i r_i$, we arrive at the kernel partial least squares model as outlined in Hoegaerts et al. (2004). In the specific case where $\psi = \phi$, its KKT conditions align with those of the LS-SVM setting for ridge regression (Saunders et al., 1998;

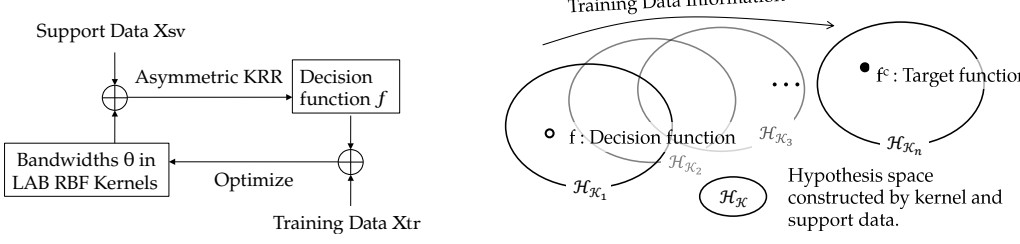

(a) Flowchart of how we learn kernels.  (b) Changes in function spaces when learning kernels.

Figure 2: The idea of the kernel learning in this paper. When interpolating support data to obtain decision function, we actually search in the hypothesis space $\mathcal{H}_{\mathcal{K}}$. When optimizing $\Theta$ with training data, we actually adapt the hypothesis space. By repeating these two operation, we finally obtain a good hypothesis space close to the target function.

Suykens et al., 2002). Furthermore, under the same condition of $\psi = \phi$ and when the regularization parameter is set to zero, it reduces to ordinary least squares regression (Hoegaerts et al., 2005).

With the aid of error variables $e$ and $r$, a clearer perspective on the relationship between $f_1$ and $f_2$ emerges. As clarified in Theorem 3, the approximation error on training data is equal to the value of dual variables, a computation facilitated through the kernel trick. Consequently, this reveals that $f_1$ and $f_2$ typically diverge when $\phi$ and $\psi$ are not equal, as they exhibit distinct approximation errors. A complementary geometric insight arises from the term $\sum_{i=1}^{N} e_i r_i$ within the objective function. This signifies that, in practice, $f_1$ and $f_2$ tend to approach the target $y$ from opposite directions because the signs in their approximation errors tend to be dissimilar. For practical applications, one may opt for the regression function with the smaller approximation error.

## 3 LEARNING LOCALLY-ADAPTIVE-BANDWIDTH RBF KERNELS

Based on the above theoretical result of asymmetric KRR, in this section we provide a learning algorithm of the LAB RBF kernel, to determine the local bandwidths $\theta_i$ and the decision function $f$. The key of our approach is that, we use a part of available data as *support data*, to which the proposed asymmetric KRR model is applied to obtained the formulation of $f$. Then we use the rest of data to train bandwidths $\theta_i$ corresponding to support data.

The training scheme is given in Fig. 2 (a): Assume a support dataset $\mathcal{Z}_{sv} = \{\mathcal{X}_{sv}, \mathcal{Y}_{sv}\}$ and a training dataset $\mathcal{Z}_{tr} = \{\mathcal{X}_{tr}, \mathcal{Y}_{tr}\}$ are pre-given. Let $\Theta$ denotes the set of bandwidths for support data, i.e., $\Theta = \{\theta_1, \cdots, \theta_{N_{sv}}\}$. We firstly fix $\Theta$ and apply asymmetric KRR model on support data to obtain the decision function, denoted by $f_{\mathcal{Z}_{sv}, \Theta}$. Denote the test data as $t$, Theorem 1 and (6) provides two choices of regression functions associated with $\boldsymbol{K}(t, \boldsymbol{X}_{sv})$ and $\boldsymbol{K}^\top(\boldsymbol{X}_{sv}, t)$, respectively. It is important to note that within LAB RBF kernels, the matrices $\boldsymbol{K}(t, \boldsymbol{X}_{sv})$ and $\boldsymbol{K}^\top(\boldsymbol{X}_{sv}, t)$ are distinct. The bandwidth of the former is dependent on $\boldsymbol{X}_{sv}$, while the bandwidth of the latter depends on $t$. Since only bandwidths for support data are optimized, we can compute solely $\boldsymbol{K}(t, \boldsymbol{X}_{sv})$, and we are unable to calculate $\boldsymbol{K}(\boldsymbol{X}_{sv}, t)$ due to the absence of bandwidth information for testing data. Consequently, only $f_1$ in (6) can be utilized to interpolate the support data. That is,

$$f_{\mathcal{Z}_{sv}, \Theta}(t) = \boldsymbol{K}_\Theta(t, \boldsymbol{X}_{sv})(\boldsymbol{K}_\Theta(\boldsymbol{X}_{sv}, \boldsymbol{X}_{sv}) + \lambda \boldsymbol{I}_N)^{-1} \boldsymbol{Y}_{sv}. \tag{8}$$

As a function that interpolates a small dataset, it is evident that the generalization performance of $f_{\mathcal{Z}_{sv}, \Theta}$ falls short of expectations. However, by introducing trainable bandwidths $\Theta$, we can substantially improve the generalization capacity of $f_{\mathcal{Z}_{sv}, \Theta}$. This enhancement is achieved by the following optimization model to train $\Theta$, enabling the function to approximate the training data:

$$\begin{aligned} \Theta^* &= \arg\min_{\Theta} \|f_{\mathcal{Z}_{sv}, \Theta}(\boldsymbol{X}_{tr}) - \boldsymbol{Y}_{tr}\|_2^2 \\ &= \arg\min_{\Theta} \|\boldsymbol{K}_\Theta(\boldsymbol{X}_{tr}, \boldsymbol{X}_{sv})(\boldsymbol{K}_\Theta(\boldsymbol{X}_{sv}, \boldsymbol{X}_{sv}) + \lambda \boldsymbol{I}_N)^{-1} \boldsymbol{Y}_{sv} - \boldsymbol{Y}_{tr}\|_2^2. \end{aligned} \tag{9}$$

After the above optimization, we finally obtain the regression function $f_{\mathcal{Z}_{sv}, \Theta^*}(t)$.

---

**Algorithm 1** Learning LAB RBF kernels with SGD and dynamic strategy.

1: **Input:** Data $\mathcal{Z} = \{\mathcal{X}, \mathcal{Y}\}$, regularization hyper-parameter $\lambda$.
2: Initialization: Error tolerance $\epsilon > 0$, initial bandwidth $\Theta^{(0)} > 0$, learning rate for gradient descent method $\eta > 0$, $k$ for the dynamic strategy, and uniformly sampled support dataset $\mathcal{Z}^{(0)} = \{\mathcal{X}_{sv}^{(0)}, \mathcal{Y}_{sv}^{(0)}\}$.
3: **repeat**
4:     Compute the function $f_{\mathcal{Z}_{sv}^{(t)}, \Theta^{(t)}}$ according to (1) and (8).
5:     t=0.
6:     **repeat**
7:         Randomly sample a subset $\{\mathcal{X}_s, \mathcal{Y}_s\} \subset \mathcal{Z} \setminus \mathcal{Z}_{sv}$.
8:         Compute $\Theta^{(t+1)} = \Theta^{(t)} + \eta \frac{\partial}{\partial \Theta} \mathcal{L}(f_{\mathcal{Z}_{sv}^{(t)}, \Theta^{(t)}}(\boldsymbol{X}_s), \boldsymbol{Y}_s)$ according to (9).
9:         t=t+1.
10:     **until** the maximal number of iteration is exceeded.
11:     Compute error $\xi_i = (f_{\mathcal{Z}_{sv}^{(t)}, \Theta^{(t)}}(\boldsymbol{x}_i) - y_i)^2$ for all data $\{\boldsymbol{x}_i, y_i\} \in \mathcal{Z} \setminus \mathcal{Z}_{sv}$.
12:     **if** $\max_i \xi_i \le \epsilon$ or the maximal support data number is exceeded, **then**
13:         break.
14:     **else**
15:         Add the first $k$ samples with largest error to the support dataset and obtain $\mathcal{Z}_{sv}^{(t+1)}$
16:     **end if**
17: **until** the maximal number of iteration is exceeded.
18: Compute the $\boldsymbol{\alpha} = (\boldsymbol{K}_{\Theta^{(t)}}(\boldsymbol{X}_{sv}^{(t)}, \boldsymbol{X}_{sv}^{(t)}) + \lambda \boldsymbol{I}_N)^{-1} \boldsymbol{Y}_{sv}^{(t)}$.
19: **Return** $\boldsymbol{\alpha}, \mathcal{Z}_{sv}^{(t)}$ and $\Theta^{(t)}$.

---

In this approach, we utilize support data in $\mathcal{Z}sv$ for optimizing the regressor and training data in $\mathcal{Z}tr$ for optimizing $\Theta$. This sets our algorithm apart from existing deep kernel learning and multiple kernel learning methods, where both the decision function and the kernel are optimized concurrently using the same data. It is worth noting that $\mathcal{Z}_{sv}$ in our strategy serves a similar role to the selected subset of training data in Nyström approximation, (see Williams & Seeger (2000); Rudi et al. (2017) for reference). However, with the integration of locally-adaptive bandwidths, our subsequent bandwidth training process significantly enhances the generalization ability of decision models, as we will demonstrate in the next experiment section. Overall, the proposed kernel learning algorithm yields advantages in terms of efficiency and effectiveness:

**Computational efficiency with small support data.** In traditional kernel-based learning, computing the kernel matrix and its inverse is the most computationally demanding operation. However, our approach confines this computation to a small dataset $\mathcal{X}_{sv}$, resulting in a manageable computational complexity of $O(N_{sv}^3)$. Moreover, our approach explores a completely different mechanism than current accelerating techniques for kernel machines. This opens the possibility of incorporating methods from, for example, Rudi et al. (2017); Abedsoltan et al. (2023) into our framework, resulting in higher computational efficiency. Additionally, in Equation (9), the relationship between the objective function and $\Theta$ is analytically determined, as $f_{\mathcal{Z}sv, \Theta}$ is explicitly presented. This enables the utilization of gradient-descent-based (GD-base) methods for training $\Theta$, leveraging advancements in hardware and software that have significantly improved computational efficiency.

**Enhanced generalization with large training data.** The training procedure for optimizing $\Theta$ in our approach closely resembles a parameter-tuning step, akin to cross-validation, commonly used to enhance the generalization ability of decision functions in traditional learning frameworks. Here the generalization ability of $f_{\mathcal{Z}_{sv}, \Theta}$ is also enhanced by effectively approximating a large amount of training data. This augmentation of generalization stems from the incorporation of training data information into the local bandwidths of LAB RBF kernels. These bandwidth adjustments essentially correspond to different hypothesis spaces, as depicted in Fig. 2 (b). By adapting these functional spaces optimally, our approach empowers the regression function to efficiently approximate larger datasets with reduced complexity compared to classical kernel-based models.

**Dynamic strategy for selecting support data.** The selection of support data is an important step in constructing a regressor, as it can significantly impact the performance of the final model. There are various strategies for selecting support data, ranging from simple random selection to more

Table 1: $R^2$ ($\uparrow$) of different regression methods on real datasets.

| Dataset | Tecator N=240,M=122 | Yacht N=308,M=6 | Airfoil N=1503,M=5 | SML N=4137,M=22 | Parkinson N=5875,M=20 | Comp-active[a] N=8192,M=21 |
|---|---|---|---|---|---|---|
| RBF KRR | 0.9586±0.0071 | 0.9889±0.0025 | 0.8634±0.0248 | 0.9779±0.0013 | 0.8919±0.0091 | 0.9822 |
| TL1 KRR | 0.9670±0.0113 | 0.9705±0.0033 | 0.9464±0.0065 | 0.9947±0.0005 | 0.9475±0.0034 | 0.9801 |
| R-SVR-MKL | 0.9711±0.0212 | 0.9945±0.0008 | 0.9201±0.01099 | 0.9959±0.0006 | 0.9032±0.0122 | 0.9834 |
| SVR-MKL | 0.9698±0.0157 | 0.9957±0.0022 | 0.9535±0.0042 | 0.9970± 0.0006 | 0.9011±0.0110 | 0.9829 |
| EigenPro3.0 | 0.9758±0.0029 | 0.9944±0.0036 | 0.9262±0.0166 | 0.9934±0.0009 | 0.9260±0.0079 | 0.9830 |
| RFMs | 0.9811±0.0078 | 0.9947±0.0018 | 0.9394 ±0.0079 | 0.9960±0.0007 | 0.9988±0.0004 | **0.9852** |
| Falkon | 0.9769±0.0086 | 0.9982±0.0024 | 0.9377 ±0.0067 | 0.9960±0.0007 | 0.9492±0.0063 | 0.9808 |
| ResNet | 0.9841±0.0067 | 0.9940±0.0003 | 0.9538±0.0066 | 0.9976±0.0004 | 0.9906±0.0048 | 0.9836 |
| WNN | **0.9875±0.0044** | 0.9924±0.0025 | 0.9128±0.0089 | 0.9926±0.0008 | 0.9139±0.0055 | 0.9817 |
| LAB RBF | 0.9752±0.0139 | **0.9985±0.0004** | **0.9608±0.0079** | **0.9990±0.00005** | **0.9990±0.0006** | 0.9835 |

[a] The test set of Comp-activ is pre-given.
Notations N, M denote the data number and the feature dimension, respectively.

sophisticated methods based on data information. For example, sorting data according to their labels and then evenly selecting a required amount of data is a reasonable and practical approach. Besides, we propose a *dynamic strategy* for selecting support data. Initially, we uniformly select $N_0$ support data points $\mathcal{Z}_{sv}^{(0)}$ and then: (i) Optimize (8) and (9) accordingly to obtain $f_{\mathcal{Z}_{sv}^{(0)},\Theta}$. (ii) Compute approximation error $(f_{\mathcal{Z}_{sv}^{(0)},\Theta}(\boldsymbol{x}_i) - y_i)^2, \forall\{\boldsymbol{x}_i, y_i\} \in \mathcal{Z}_{tr}$. (iii) Add data with first $k$ largest error to form a new support dataset $\mathcal{Z}_{sv}^{(1)}$. Repeat the above process until all approximation error is less than a pre-given threshold or the maximal support data number is exceeded.

In Alg. 1, the overall algorithm is presented with dynamic strategy and stochastic gradient descent (SGD) methods. Therefore, the convergence analysis for the optimization (9) in the algorithm can adopt the current framework and results from SGD for non-convex functions. For more detailed information, please refer to Fehrman et al. (2020). Additionally, the use of SGD requires initialization of $\Theta$, and in our experiments, we suggest using the global bandwidth of a general RBF kernel, tuned on the training data, as the initial parameter.

## 4 NUMERICAL EXPERIMENTS

This section is dedicated to assessing the performance of the proposed LAB RBF kernels and Alg. 1 by comparing them to widely used regression methods on real datasets. We particularly focus on addressing the following questions: *(1) Does training bandwidths on the training dataset significantly enhance generalization ability? (2) Can the utilization of LAB RBF kernels lead to a reduction in the number of support data and make them applicable to large-scale datasets?*

### 4.1 EXPERIMENT SETTING

**Datasets.** Real datasets include: Yacht, Airfoil, Parkinson (Tsanas et al., 2009), SML, Electrical, Tomshardware from UCI dataset[2] (Asuncion & Newman, 2007), Tecator from StatLib [3] (Vlachos & Meyer, 2005),Comp-active from Toronto University[4], and KC House from Kaggle [5] (Harlfoxem, 2016). The detailed description of datasets are provided in Appendix C. Each feature dimension of data and the label are normalized to $[-1, 1]$.

**Measurement.** We use R-squared ($R^2$), also known as the coefficient of determination (refer to Gelman et al. (2019) for more details), on the test set $\mathcal{Z}_{test}$ to evaluate the regression performance.

$$R^2 = 1 - \frac{\sum_{(\boldsymbol{x}_i, y_i) \in \mathcal{Z}_{test}} (y_i - \hat{f}(\boldsymbol{x}_i))^2}{\sum_{(\boldsymbol{x}_i, y_i) \in \mathcal{Z}_{test}} (y_i - \bar{y})^2},$$

where $\hat{f}$ is the estimated function, and $\bar{y}$ is the mean of labels. All the following experiments randomly take $80\%$ of the total data as training data and the rest as testing data, and are repeated 50 times.

[2] https://archive.ics.uci.edu/ml/datasets.php
[3] http://lib.stat.cmu.edu/datasets/
[4] https://www.cs.toronto.edu/~delve/data/comp-activ/desc.html
[5] https://www.kaggle.com/datasets/harlfoxem/housesalesprediction

Table 2: Number of support vectors of different kernel-based regression methods on real datasets.

| Dataset | Tecator | Yacht | Airfoil | SML | Parkinson | Comp-active |
|---------|---------|-------|---------|------|-----------|-------------|
| R-SVR-MKL | 174.2 | 224.7 | 953.1 | 2844 | 4047 | 1397 |
| SVR-MKL | 160.7 | 144.5 | 1035 | 1424 | 3759 | 1423 |
| Falkon | 100 | 200 | 900 | 2000 | 4000 | 1500 |
| EigenPro3.0 | 192 | 247 | 1203 | 3310 | 4700 | 6554 |
| LAB RBF | 20 | 30 | 200 | 340 | 350 | 70 |

All the experiments were conducted using Python on a computer equipped with an AMD Ryzen 9 5950X 16-Core 3.40 GHz processor, 64GB RAM, and an NVIDIA GeForce RTX 4060 GPU with 8GB memory. The code will be made publicly accessible after the review process.

**Compared methods.** We compared nine regression methods, including two traditional kernel regression methods using RBF (RBF KRR, (Vovk, 2013)) and indefinite TL1 kernels (TL1 KRR, (Huang et al., 2018)). Additionally, there are multiple kernel learning methods applied on support vector regression, denoted as SVR-MKL and R-SVR-MKL (using only RBF kernel candidates). We also consider three recent kernel methods: Falkon (Rudi et al., 2017; Meanti et al., 2022), EigenPro3.0 (Abedsoltan et al., 2023), Recursive feature machines (RFMs, (Radhakrishnan et al., 2022)), with the first two being based on the Nyström method. Finally, two neural network-based methods are included: ResNet (Chen et al., 2020), and wide neural network (WNN).

All setting and hyper-parameters of these methods are determined for each dataset by 5-fold cross-validation. Details of compared methods and hyper-parameters are given in Appendix D.

## 4.2 EXPERIMENTAL RESULTS

**Compared with popular regression methods.** The results of the regression analysis on small-scale datasets, as measured by the $R^2$, are presented in Table 1. It is evident that greater model flexibility leads to improved regression accuracy, thus highlighting the benefits of flexible models. Notably, TL1 KRR outperforms RBF KRR in most datasets due to its indefinite nature. R-SVR-MKL, which considers a larger number of RBF kernels, exhibits much better performance than RBF KRR. While SVR-MKL, which considers a wider range of kernel types, achieves even higher accuracy compared to R-SVR-MKL. Among the neural network models, both ResNet and WNN demonstrate superior performance to the aforementioned methods. Advanced kernel methods, including Falkon, EigenPro3.0, and RFMs, also present significant improvement over traditionay kernel methods. Overall, our proposed LAB RBF achieves the highest regression accuracy, significantly increasing the $R^2$ compared to the baseline. Notably, LAB RBF performs better than ResNet in certain datasets, indicating that LAB RBF kernels offer sufficient flexibility and training bandwidths on the training dataset is indeed effective to enhance the model generalization ability.

Table 2 additionally reports the number of support vectors of sparse kernel methods, which enables a more intuitive understanding of the sizes of decision models. Specifically, it provides the maximal support data number in Alg. 1 for LAB RBF, and the predefined number of centers for Falkon, and the average number of support vectors for SVR methods. It should be noted that KRR uses all training data as support data, which results in a much larger complexity of the decision model compared to other kernel methods. This observation further underscores the advantage of enhancing kernel flexibility and learning kernels, as demonstrated by the compact size of the decision model achieved with our proposed LAB RBF kernel.

**Performance on large-scale datasets.** Traditional kernel-based algorithms are inefficient on large-scale datasets due to the matrix inverse operator on the large kernel matrix. Consequently, we compare our algorithm with three advanced kernel methods and two neural-network-based methods. The results are presented in Table 3, which more prominently underscores the capability of LAB RBF kernels in effectively reducing the required number of support data. Furthermore, it achieves a comparable level of regression accuracy to other advanced choices designed for such datasets. Notably, these advanced methods exhibit substantial model sizes. For instance, ResNet has a substantial number of parameters, and RFMs utilize all of the training data as support data. Although Falkon and EigenPro3.0 are based on the Nyström method, their reliance on symmetric kernel functions forces them to use a large amount of training data as support data to achieve high accuracy. In contrast, LAB RBF kernels

Table 3: Mean and standard of $R^2(\uparrow)$ of different algorithms in large-scale datasets.

| Dataset | Electrical | KC House | TomsHardware |
|---|---|---|---|
| | N=10000,M= 11 | N=21623,M=14 | N=28179,M=96 |
| WNN | 0.9617±0.0027 | 0.8501±0.0194 | 0.9248±0.0303 |
| ResNet | **0.9705±0.0025** | 0.8823±0.0117 | 0.9697±0.0021 |
| EigenPro3.0 (#S.V.)[a] | 0.9513±0.0024(8000) | 0.8636±0.0119(17291) | 0.9436±0.0282 (20000) |
| RFMs (#S.V.) | 0.9582±0.0028(8000) | 0.9008±0.0072 (17291) | 0.9115±0.0031(22544) |
| Falkon (#S.V.) | 0.9532±0.0025 (3000) | 0.8640±0.0145 (5000) | 0.9001±0.0143 (5000) |
| LAB RBF kernels (#S.V.) | 0.9642±0.0021 (300) | **0.8917±0.0086**(400) | **0.9809±0.0028**(500) |

[a] Notation # S.V. denotes the number of support vectors.

maintain a comparatively low number of support data, attributed to the high flexibility provided by locally adaptive bandwidths and the kernel learning algorithm.

## 5 RELATED WORKS

**RBF kernels with diverse bandwidths.** Long-standing research in statistics, particularly in kernel regression and kernel density estimation, such as Abramson (1982); Brockmann et al. (1993); Zheng et al. (2013), has investigated RBF kernels with varying bandwidths in local regions. These studies have consistently shown the superiority of locally adaptive bandwidth estimators over global estimators in theory and simulations. However, due to computational constraints and specific problem settings, these analyses have largely focused on one-dimensional algorithms without addressing generalization. In the field of machine learning, works like automatic relevance determination Neal (2012) and hierarchical Gaussian kernels (Steinwart et al., 2016; Hang & Steinwart, 2021) propose to employ heterogeneous bandwidths specific to each feature. Recent research in Radhakrishnan et al. (2022) integrates the concept of deep feature learning into kernel learning through efficient algorithms, albeit confined to the domain of symmetric kernels. There has been a limited exploration of data-dependent bandwidths, largely due to a lack of understanding and application of asymmetric kernels. In this paper, we address this limitation by introducing an innovative asymmetric KRR framework. Consequently, we achieve locally data-adaptive bandwidths tailored to RBF kernels.

**Asymmetric kernel-based learning.** Existing research in asymmetric kernel learning has primarily proposed frameworks based on SVD (Suykens, 2016) and least square SVM (He et al., 2023), lacking interpretable models for other tasks. Furthermore, these works predominantly concentrate on the application of asymmetric kernels within established kernel methods (Koide & Yamashita, 2006) and the interpretation of associated optimization models (Lin et al., 2022). Despite notable progress in these contexts, the field still grapples with the scarcity of versatile asymmetric kernel functions. Current applications of asymmetric kernel matrices often rely on datasets (e.g. the directed graph in He et al. (2023)) or recognized asymmetric similarity measures (e.g. the Kullback-Leibler kernels in Moreno et al. (2003)) This yields improved performance in specific scenarios but leaving a significant gap in addressing diverse datasets. With the help of trainable LAB RBF kernels, the asymmetric kernel learning framework proposed in this paper thus lays a robust groundwork for utilizing asymmetric kernels in tackling general regression tasks.

## 6 CONCLUSION

This paper introduces locally-adaptive-bandwidths to RBF kernels, significantly enhancing the flexibility of the resulting LAB RBF kernels. We tackle the inherent asymmetry of LAB RBF kernels by establishing an asymmetric KRR framework, demonstrating that one of its stationary points maintains a structure akin to classical symmetric KRR. Additionally, an efficient LAB RBF kernel learning algorithm is devised, allowing for bandwidth determination with control over support vectors. The experimental results underscore the algorithm's superiority in reducing the required number of support data, surpassing state-of-the-art accuracy, and even outperforming well-trained residual networks. This work underscores the benefits of enhancing kernel flexibility and highlights the effectiveness of the proposed asymmetric kernel learning approach. Considering the fundamental role of non-Mercer kernels and asymmetric kernels in modern deep learning architectures, such as transformers (Wright & Gonzalez, 2021; Chen et al., 2023), our GD-based asymmetric kernel learning algorithm exhibits great potential for future integration with deep architectures.

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
