## APPENDIX A  PROOF OF THEOREM 1

Here we present the proof of Theorem 1.

*Proof.* Based on Equation (3), we can express $\boldsymbol{w}^*$ as:

$$\boldsymbol{w}^* = \psi(\boldsymbol{X})(\phi^\top(\boldsymbol{X})\psi(\boldsymbol{X}) + \lambda \boldsymbol{I}_N)^{-1}\boldsymbol{Y} \overset{(a)}{=} (\lambda \boldsymbol{I}_F + \psi(\boldsymbol{X})\phi^\top(\boldsymbol{X}))^{-1}\psi(\boldsymbol{X})\boldsymbol{Y} \quad (10)$$

where equation (a) is derived from (3) with $\boldsymbol{A} = \boldsymbol{I}_F$, $\boldsymbol{B} = \psi(\boldsymbol{X})$, $\boldsymbol{C} = \phi^\top(\boldsymbol{X})$, $\boldsymbol{D} = \boldsymbol{I}_N$. Similarly, for $\boldsymbol{v}^*$, we have

$$\boldsymbol{v}^* = \phi(\boldsymbol{X})(\psi^\top(\boldsymbol{X})\phi(\boldsymbol{X}) + \lambda \boldsymbol{I}_N)^{-1}\boldsymbol{Y} \overset{(b)}{=} (\lambda \boldsymbol{I}_F + \phi(\boldsymbol{X})\psi^\top(\boldsymbol{X}))^{-1}\phi(\boldsymbol{X})\boldsymbol{Y}, \quad (11)$$

where equation (b) again applies (3) with $\boldsymbol{A} = \boldsymbol{I}_F$, $\boldsymbol{B} = \phi(\boldsymbol{X})$, $\boldsymbol{C} = \psi^\top(\boldsymbol{X})$, $\boldsymbol{D} = \boldsymbol{I}_N$. Take the derivation of the objective function with respect to $\boldsymbol{w}$ and $\boldsymbol{v}$ at point $(\boldsymbol{w}^*, \boldsymbol{v}^*)$, we observe:

$$\frac{\partial L}{\partial \boldsymbol{w}}\Big|_{\substack{\boldsymbol{w}=\boldsymbol{w}^* \\ \boldsymbol{v}=\boldsymbol{v}^*}} = (\lambda \boldsymbol{I}_F + \phi(\boldsymbol{X})\psi^\top(\boldsymbol{X}))(\lambda \boldsymbol{I}_F + \phi(\boldsymbol{X})\psi^\top(\boldsymbol{X}))^{-1}\phi(\boldsymbol{X})\boldsymbol{Y} - \phi(\boldsymbol{X})\boldsymbol{Y} = 0,$$

$$\frac{\partial L}{\partial \boldsymbol{v}}\Big|_{\substack{\boldsymbol{w}=\boldsymbol{w}^* \\ \boldsymbol{v}=\boldsymbol{v}^*}} = (\lambda \boldsymbol{I}_F + \psi(\boldsymbol{X})\phi^\top(\boldsymbol{X}))(\lambda \boldsymbol{I}_F + \psi(\boldsymbol{X})\phi^\top(\boldsymbol{X}))^{-1}\psi(\boldsymbol{X})\boldsymbol{Y} - \psi(\boldsymbol{X})\boldsymbol{Y} = 0.$$

This verifies that the point $(\boldsymbol{w}^*, \boldsymbol{v}^*)$ satisfies the stationarity condition. $\square$

## APPENDIX B  PROOF OF THEOREM 3

Here we present the proof of Theorem 3.

*Proof.* The Lagrangian of (7) is

$$\mathcal{L} = \lambda \boldsymbol{w}^\top \boldsymbol{v} + \sum_{i=1}^N e_i r_i + \sum_i \alpha_i(y_i - e_i - \phi(\boldsymbol{x}_i)^\top \boldsymbol{w}) + \sum_i \beta_i(y_i - r_i - \psi(\boldsymbol{x}_i)^\top \boldsymbol{v}), \quad (12)$$

where $\boldsymbol{\alpha} \in \mathcal{R}^N$ and $\boldsymbol{\beta} \in \mathcal{R}^N$ are Lagrange multipliers. The KKT conditions lead to

$$\frac{\partial \mathcal{L}}{\partial \boldsymbol{v}} = \lambda \boldsymbol{w} - \psi(\boldsymbol{X})\boldsymbol{\beta} = 0 \qquad \Longrightarrow \boldsymbol{w} = \frac{1}{\lambda}\psi(\boldsymbol{X})\boldsymbol{\beta},$$

$$\frac{\partial \mathcal{L}}{\partial \boldsymbol{w}} = \lambda \boldsymbol{v} - \phi(\boldsymbol{X})\boldsymbol{\alpha} = 0 \qquad \Longrightarrow \boldsymbol{v} = \frac{1}{\lambda}\phi(\boldsymbol{X})\boldsymbol{\alpha},$$

$$\frac{\partial \mathcal{L}}{\partial r_i} = e_i - \beta_i = 0 \qquad \Longrightarrow e_i = \beta_i,$$

$$\frac{\partial \mathcal{L}}{\partial e_i} = r_i - \alpha_i = 0 \qquad \Longrightarrow r_i = \alpha_i,$$

$$\frac{\partial \mathcal{L}}{\partial \alpha_i} = y_i - e_i - \phi(\boldsymbol{x}_i)^\top \boldsymbol{w} = 0 \qquad \Longrightarrow e_i = y_i - \phi(\boldsymbol{x}_i)^\top \boldsymbol{w},$$

$$\frac{\partial \mathcal{L}}{\partial \beta_i} = y_i - r_i - \psi(\boldsymbol{x}_i)^\top \boldsymbol{v} = 0 \qquad \Longrightarrow r_i = y_i - \psi(\boldsymbol{x}_i)^\top \boldsymbol{v}.$$

Substitute the first four lines into the last two lines, we can eliminate primal variables $\boldsymbol{w}, \boldsymbol{v}, \boldsymbol{e}, \boldsymbol{r}$:

$$\boldsymbol{\beta}^* = \boldsymbol{Y} - \frac{1}{\lambda}\phi(\boldsymbol{X})^\top \psi(\boldsymbol{X})\boldsymbol{\beta}^* \qquad \Longrightarrow \boldsymbol{\beta}^* = \lambda(\lambda \boldsymbol{I}_N + \phi(\boldsymbol{X})^\top \psi(\boldsymbol{X}))^{-1}\boldsymbol{Y},$$

$$\boldsymbol{\alpha}^* = \boldsymbol{Y} - \frac{1}{\lambda}\psi(\boldsymbol{X})^\top \phi(\boldsymbol{X})\boldsymbol{\alpha}^* \qquad \Longrightarrow \boldsymbol{\alpha}^* = \lambda(\lambda \boldsymbol{I}_N + \psi(\boldsymbol{X})^\top \phi(\boldsymbol{X}))^{-1}\boldsymbol{Y}.$$

Thus, we get the result in Theorem 3 and the proof is completed. $\square$

APPENDIX C    DETAILS OF DATASETS AND THE HYPER-PARAMETER SETTING.

**Tecator**: The objective is to predict the fat content of a meat sample based on its near infrared absorbance spectrum. These measurements are obtained using a Tecator Infratec Food and Feed Analyzer operating in the wavelength range of 850 - 1050 nm through the Near Infrared Transmission (NIT) principle. The dataset can be download from `http://lib.stat.cmu.edu/datasets/tecator`.

**Yacht**: The Yacht Hydrodynamics Data Set aims to predict the residuary resistance of sailing yachts based on features such as fundamental hull dimensions and boat velocity. The dataset consists of 308 full-scale experiments conducted at the Delft Ship Hydromechanics Laboratory for this purpose. The dataset can be download from `https://archive.ics.uci.edu/dataset/243/yacht+hydrodynamics`.

**Airfoil**: The Airfoil Self-Noise dataset is sourced from NASA and encompasses aerodynamic and acoustic tests on two and three-dimensional airfoil blade sections carried out in an anechoic wind tunnel. The objective is to predict the scaled sound pressure level. The dataset can be download from `https://archive.ics.uci.edu/dataset/291/airfoil+self+noise`.

**SML**: The SML2010 dataset is collected from a monitoring system installed in a domotic house and covers approximately 40 days of monitoring data. The dataset contains missing values, which were imputed using the mean value. The dataset can be download from `https://archive.ics.uci.edu/dataset/274/sml2010`.

**Parkinson**: The Oxford Parkinson's Disease Telemonitoring Dataset comprises various biomedical voice measurements from 42 individuals with early-stage Parkinson's disease enrolled in a six-month trial of a telemonitoring device for remote symptom progression monitoring. The dataset can be download from `https://archive.ics.uci.edu/dataset/189/parkinsons+telemonitoring`.

**Comp-activ**: The ComputerActivity database records diverse performance metrics, such as bytes read/written from system memory, from a Sun Sparctation 20/712 with 2 CPUs and 128 MB of main memory. The dataset can be download from `https://www.cs.toronto.edu/~delve/data/comp-activ/desc.html`.

**TomsHardware**: This dataset is a part of Buzz in social media data set, containing examples of buzz events from the social network Tom's Hardware. The dataset can be download from `https://archive.ics.uci.edu/dataset/248/buzz+in+social+media`.

**KC House**: The KC House dataset focuses on house prices in King County, encompassing Seattle, and includes homes sold between May 2014 and May 2015. The dataset can be download from `https://www.kaggle.com/datasets/shivachandel/kc-house-data`.

**Electrical**: The Electrical dataset pertains to the local stability analysis of a 4-node star system, where the electricity producer is situated at the center. This system implements the Decentral Smart Grid Control concept. The dataset can be download from `https://archive.ics.uci.edu/dataset/471/electrical+grid+stability+simulated+data`.

APPENDIX D    DETAILS OF COMPARED METHODS AND HYPER-PARAMETER SETTING.

**Compared methods: Compared methods.** Six regression methods are compared in this experiment, including:

- RBF KRR (Vovk, 2013): classical kernel ridge regression with conventional RBF kernels, served as the baseline.

- TL1 KRR: classical kernel ridge regression employing an indefinite kernel named Truncated $\ell_1$ kernel (Huang et al., 2018). The expression of TL1 kernel is $\mathcal{K}(\boldsymbol{x}, \boldsymbol{x}') = \max\{\rho - \|\boldsymbol{x} - \boldsymbol{x}'\|_1, 0\}$, where $\rho > 0$ is a pre-given hyper-parameter. The TL1 kernel is a piecewise linear indefinite kernel and is expected to be more flexible and have better performance than the conventional RBF kernel.

- SVR-MKL: Multiple kernel learning applied on support vector regression. The kernel dictionary includes RBF kernels, Laplace kernels, and polynomial kernels. Results for R-SVR-MKL with only RBF kernels are also provided. The implementation of MKL is available in the Python package MKLpy (Aiolli & Donini, 2015; Lauriola & Aiolli, 2020).

- Falkon (Rudi et al., 2017; Meanti et al., 2022): An advanced and well-developed algorithm for KRR that employs hyper-parameter tuning techniques to enhance accuracy and utilizes Nyström approximation to reduce the number of support data points, enabling it to handle large-scale datasets. We used the public code of Falkon, available at `https://github.com/FalkonML/falkon`.

- EigenPro3.0 (Abedsoltan et al., 2023): An advanced general kernel machine for large datasets, utilizing Nystöm methods and projected dual preconditioned SGD. We used the public code of EigenPro3.0, available at `https://github.com/EigenPro/EigenPro3`.

- RFMs (Radhakrishnan et al., 2022): Recursive feature machines is advanced kernel methods which utilizes the mechanism of deep feature learning, resulting high efficient algorithms and ability to handle large datasets. We used the public code of RFMs, available at `https://github.com/aradha/recursive_feature_machines`.

- ResNet: The regression version of ResNet follows the structure in Chen et al. (2020), and the code is available in `https://github.com/DowellChan/ResNetRegression`.

- WNN: The regression version of a wide neural network, which is fully-connected and has only one hidden layer.

**Implementation details.** Among the compared methods, Kernel Ridge Regression (KRR) stands as the fundamental technique that combines the Tikhonov regularized model with the kernel trick. The coefficients of kernels for both SVR-MKL and R-SVR-MKL are calculated following the approach in EasyMKL (Aiolli & Donini, 2015). For Falkon, the code is available at `https://github.com/FalkonML/falkon`. LAB RBF, ResNet, and WNN are optimized using gradient methods with varying hyper-parameters such as initial points, learning rate, and batch size. The initial weights of both ResNet and WNN are set according to the Kaiming initialization introduced in He et al. (2015). In the subsequent experiments, the Adam optimizer is initially used, and upon stopping, the SGD optimizer is applied. Early stopping is implemented for the training of ResNet and WNN, where 10% of the training data is sampled to form a validation set, and validation loss is assessed every epoch. The epoch with the best validation loss is selected for testing. Detailed hyper-parameters of all compared methods are provided in Table 4 (for small-scale datasets) and Table 5 (for large-scale datasets).

The regression version of ResNet follows the structure in Chen et al. (2020), which has available code in `https://github.com/DowellChan/ResNetRegression`. Following the structures in Chen et al. (2020), the ResNet block has two types: Identity Block (where the dimension of input and output are the same) and Dense Block (where the dimension of input and output are different). The details of these two block are presented in Fig. 3. Considering the different dataset sizes, we use two structures of ResNet in our experiments, denoted by ResNet and ResNetSmall. For the ResNet, we use two Dense Blocks (M-W-100) and two Identity Block (100-100-100) and a linear predict layer (100-1). For the ResNetSmall, we use two Dense Blocks (M-W-50) and a linear predict layer (50-1). Here W is a pre-given width for the network.

## APPENDIX E  ADDITIONAL EXPERIMENT: IMPACT OF HYPER-PARAMETERS

In this section, we use synthetic data to evaluate the influence of hyper-parameters in Alg. 1. Synthetic data come from typical nonlinear regression test functions provided by Cherkassky et al. (1996) and take the following formulations,

$$f_1(\boldsymbol{x}) = \frac{1 + \sin(2x(1) + 3x(2))}{3.5 + \sin(x(1) - x(2))}, \ D = [-2, -2]^2,$$

$$f_2(\boldsymbol{x}) = 10\sin(\pi x(1)x(2)) + 20(x(3) - 0.5)^2 + 5x(4) + 10x(5) + 0x(6), \ D = [-1, 1]^6,$$

$$f_3(\boldsymbol{x}) = \exp(2\pi x(1)(\sin(x(4))) + \sin(x(2)x(3))), \ D = [-0.25, 0.25]^4,$$

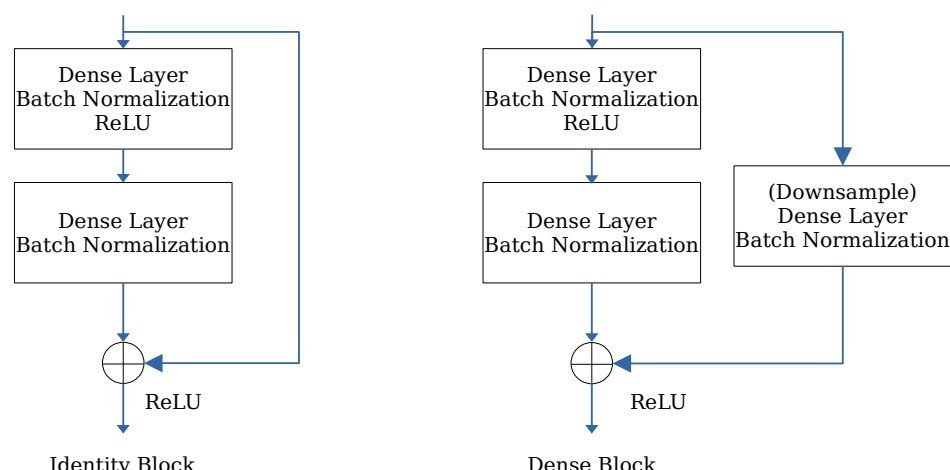

Figure 3: The structures of Identity block and Dense block.

Table 4: Hyper-parameters of eight regression methods for real datasets.

| | Hyper-parameters | Tecator | Yacht | Airfoil | SML | Parkinson | Comp_activ |
|---|---|---|---|---|---|---|---|
| LAB RBF | lr | 0.001 | 0.01 | 0.01 | 0.05 | 0.01 | 0.001 |
| | Batch size | 16 | 128 | 128 | 128 | 128 | 128 |
| | $\sigma_0$ | 0.5 | 3 | 0.2 | 50 | 30 | 0.1 |
| R-SVR-MKL | C | 1000 | 1000 | 1 | 1000 | 10 | 1 |
| | $\epsilon$ | 0.001 | 0.001 | 0.01 | 0.001 | 0.01 | 0.01 |
| | Dictionary | RBF kernels: $[100, 50, 10, 1, 0.1, 0.01, 0.001]$ | | | | | |
| SVR-MKL | C | 1000 | 1000 | 100 | 1000 | 1000 | 1000 |
| | $\epsilon$ | 0.01 | 0.01 | 0.01 | 0.01 | 0.01 | 0.01 |
| | Dictionary | RBF kernels: $[100, 1, 0.1, 0.001]$ | | | | | |
| | | Laplace kernels: $[100, 1, 0.1, 0.001]$ | | | | | |
| | | Polynomial kernels: $[1, 2, 4, 10]$ | | | | | |
| RBF KRR | $\sigma$ | 1 | 5 | 80 | 5 | 20 | 10 |
| | $\lambda$ | 0.01 | 0.001 | 0.001 | 0.01 | 0.001 | 0.001 |
| TL1 KRR | $\rho$ | 98 | 6 | 2.5 | 22 | 14 | 15 |
| | $\lambda$ | 0.001 | 0.001 | 0.001 | 0.1 | 0.01 | 0.001 |
| Falkon | $\lambda$ | 1e-6 | 1e-7 | 1e-6 | 1e-5 | 1e-7 | 1e-6 |
| | Center[a] | 100 | 200 | 900 | 2000 | 4000 | 1500 |
| | $\sigma$ | 10 | 1 | 2 | 1 | 0.7 | 2.5 |
| EigenPro3.0 | $\sigma$ | 3 | 1 | 0.5 | 1 | 0.5 | 10 |
| | Center[a] | 197 | 247 | 1203 | 3310 | 4700 | 6554 |
| ResNet | lr | 0.001 | 0.001 | 0.001 | 0.001 | 0.001 | 0.001 |
| | Batch size | 32 | 32 | 128 | 128 | 128 | 128 |
| | Structure (Width)[b] | 2(500) | 2(500) | 1(1000) | 1(1000) | 1(500) | 1(2000) |
| WNN | lr | 0.001 | 0.001 | 0.001 | 0.001 | 0.001 | 0.001 |
| | Batch size | 32 | 32 | 128 | 128 | 128 | 128 |
| | Width | 800 | 500 | 6000 | 1500 | 3000 | 9000 |

[a] The center number of Nyström approximation.
[b] Structure 1:$M - W - 100 - 100 - 100 - 1$, Structure 2: $M - W - 50 - 1$.

Table 5: Hyper-parameters of four regression methods for real datasets.

|  | Hyper-parameters | TomsHardware | Electrical | KC House |
|---|---|---|---|---|
| LAB RBF | lr | 0.001 | 0.001 | 0.001 |
|  | Batch size | 256 | 256 | 256 |
|  | $\sigma_0$ | 0.1 | 0.1 | 1 |
| Falkon | $\lambda$ | 1e-6 | 1e-6 | 1e-6 |
|  | Center | 3000 | 3000 | 5000 |
|  | $\sigma$ | 2 | 10 | 5 |
| EigenPro3.0 | $\sigma$ | 7 | 1 | 5 |
|  | Center | 20000 | 8000 | 17291 |
| ResNet | lr | 0.001 | 0.001 | 0.001 |
|  | Batch size | 128 | 256 | 256 |
|  | Structure (Width)[a] | 1(3000) | 1(2000) | 1(2000) |
| WNN | lr | 0.01 | 0.01 | 0.01 |
|  | Batch size | 128 | 256 | 128 |
|  | Width | 3000 | 3000 | 5000 |

[a] Structure 1:$M - W - 100 - 100 - 100 - 1$.

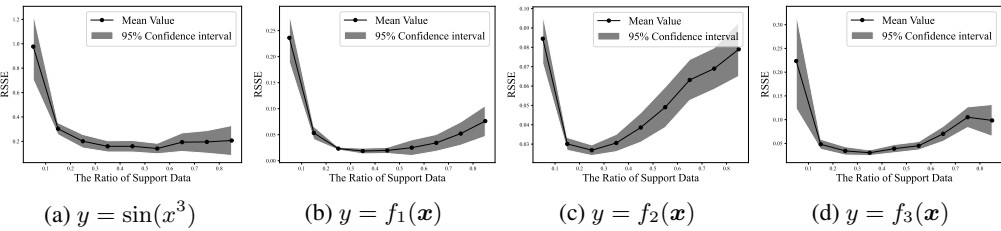

| (a) $y = \sin(x^3)$ | (b) $y = f_1(\boldsymbol{x})$ | (c) $y = f_2(\boldsymbol{x})$ | (d) $y = f_3(\boldsymbol{x})$ |

Figure 4: The mean RSSE of Alg. 1 and its standard derivation with respect to different ratio of support data. $N_{tr} = 200, N_{test} = 800$. Noise level: (a) $n = 0.4$, (b-d) $n = 0.1$.

where $D = [a, b]^n = \{\boldsymbol{x} | \boldsymbol{x} \in \mathcal{R}^n, a \le x(i) \le b, \forall 1 \le i \le n\}$. The relative sum of square error (RSSE=$1-R^2$) is reported to measure the regression performance.

**The impact of the number of support data.** The flexibility, or the number of trainable parameters, introduced by LAB RBF kernels is directly controlled by the number of support data in practice[6]. To investigate the impact of support data number on the performance of Alg. 1, we conducted experiments on synthetic datasets with varying ratios of support data, and the results are presented in Figure 4. For all functions, we performed random sampling with 200 data points allocated for training and 800 for testing. Initial bandwidths were uniformly set to $\Theta^{(0)} = 1/M$. We introduced noise at a specified level $n$, with $n$ representing the ratio of noise variance to target variance.

Our findings indicate that Alg. 1 performs optimally when the ratio of support data is in the range of $30\% - 70\%$, depending on the feature dimension. When there are more support data points, the hypothesis space gains greater capacity to capture intricate patterns. However, an excessive number of support data points can lead to overfitting because the remain training data is insufficient, where the model behaves more like simple kernel-based interpolation and becomes sensitive to noise. Conversely, too few support data points may result in underfitting, limiting the model's ability to approximate the data. Thus, selecting an appropriate number of support data points is essential to strike a balance between model complexity and the risk of overfitting.

**The impact of the initial parameter $\Theta^{(0)}$.** Here, we consider the impact of initial parameter $\Theta^{(0)}$ because generally, initial points have significant impact on both non-convex optimization and gradient descent method. Figure 5 displays the curve of RSSE with respect to different $\Theta^{(0)}$ indicating that the performance is good and stable in a wide range of $[10^{-2}, 10]$. This result further validates

---

[6]Note that there is another hyper-parameter in Alg. 1 named error tolerance to control the number of support data. But in practical implementation, setting the maximal support data number has almost equal effect as setting error tolerance, and thus only one of them is considered.

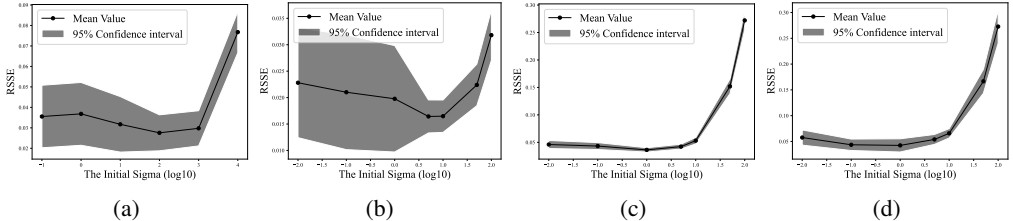

Figure 5: The mean RSSE of Alg. 1 and its standard derivation with respect to different initial $\sigma$. $N_{sv} = 100, N_{tr} = 100, N_{te} = 800$. Datasets: (a) $y = \sin(x^3)$. (b) $y = f_1(\boldsymbol{x})$. (c) $y = f_2(\boldsymbol{x})$. (d) $y = f_3(\boldsymbol{x})$. Noise level: (a) $n = 0.4$, (b-d) $n = 0.1$.

Table 6: The mean $R^2(\uparrow)$ and its standard derivation of Alg. 1 with respect to different iterations in SGD on dataset Yacht.

| Iterations | 0 | 100 | 200 | 300 | 400 |
|---|---|---|---|---|---|
| Mean of R2 | $0.5633\pm0.0737$ | $0.9396\pm0.0316$ | $0.9789\pm0.0187$ | $0.9872\pm0.0105$ | $0.9902\pm0.0073$ |
| Iterations | 500 | 1000 | 2000 | 3000 | 4000 |
| Mean of R2 | $0.9913\pm0.0065$ | $0.9940\pm0.0056$ | $0.9956\pm0.0033$ | $0.9966\pm0.0023$ | $0.9966\pm0.0024$ |

the robustness of our kernel learning algorithm compared to conventional RBF kernels, which are typically sensitive to the pre-given bandwidth.

**The impact of SGD hyper-parameters.**

Bandwidths play a crucial role in the performance of LAB RBF kernels. In our kernel learning algorithm, we use SGD methods to estimate the value of bandwidths. In practice, various factors during training, such as stopping criteria, learning rate, and batch size, may result in distinct bandwidth estimations. To evaluate the impact of these hyper-parameters on the performance of our algorithms, we conduct following experiments.

Table 6 and 7 present the performance of Alg. 1 with respect to different iteration numbers in SGD on dataset Yacht and Parkinson. Table 8 and 9 present the performance of Alg. 1 with respect to different learning rate in SGD on dataset Yacht and Parkinson. Table 10 presents the performance of Alg. 1 with respect to different batch size in SGD on dataset Yacht and Parkinson. These results underscore that the carefully selection of these hyper-parameters enhances the final performance. Nevertheless, even under sub-optimal hyper-parameter settings, the performance remains commendable, albeit with varying bandwidth estimates. This highlights the robustness and insensitivity of our algorithm across a wide spectrum of hyper-parameter choices.

## APPENDIX F  STUDY ON DIFFERENT SELECTION STRATEGY OF INITIAL SUPPORT DATA

The selection of support data has the significant influence on the performance of the proposed algorithm. In light of this, we have introduced a dynamic strategy aimed at mitigating the impact of initial support data selection in the manuscript.

Table 7: The mean $R^2(\uparrow)$ and its standard derivation of Alg. 1 with respect to different iterations in SGD on dataset Parkinson.

| Iterations | 0 | 100 | 200 | 300 | 400 |
|---|---|---|---|---|---|
| Mean of R2 | $0.5458\pm0.0315$ | $0.9101\pm0.0065$ | $0.9445\pm0.0061$ | $0.9622\pm0.0052$ | $0.9752\pm0.0061$ |
| Iterations | 500 | 1000 | 2000 | 3000 | 4000 |
| Mean of R2 | $0.9814\pm0.0055$ | $0.9889\pm0.0033$ | $0.9895\pm0.0023$ | $0.9925\pm0.0026$ | $0.9931\pm0.0019$ |

Table 8: The mean $R^2(\uparrow)$ and its standard derivation of Alg. 1 with respect to different learning rate in SGD on dataset Yacht.

| Learning Rate | 1.00E+00 | 1.00E-01 | 1.00E-02 | 1.00E-03 | 5.00E-04 | 1.00E-04 |
|---|---|---|---|---|---|---|
| Mean of R2 | 0.9723 | 0.9968 | 0.9970 | 0.9911 | 0.9804 | 0.8089 |
| Std of R2 | 0.0164 | 0.0019 | 0.0017 | 0.0038 | 0.0104 | 0.0246 |

Table 9: The mean $R^2(\uparrow)$ and its standard derivation of Alg. 1 with respect to different learning rate in SGD on dataset Parkinson.

| Learning Rate | 5.00E-01 | 1.00E-01 | 5.00E-02 | 1.00E-02 | 5.00E-03 | 1.00E-03 |
|---|---|---|---|---|---|---|
| Mean of R2 | 0.9827 | 0.9941 | 0.9943 | 0.9906 | 0.9763 | 0.8947 |
| Std of R2 | 0.0031 | 0.0018 | 0.0018 | 0.0022 | 0.0058 | 0.0052 |

In this section, we delve deeper into the effects of various methods for selecting initial support data and assess the efficacy of the introduced dynamic strategy. We will explore three different approaches to initial data selection: two rational methods (Y-based and X-based) and one irrational method (Extreme Y).

- Y-based (utilized in the manuscript): data is sorted based on their labels, and support data is uniformly selected.
- X-based: k-means is applied to the training data to identify cluster centers, followed by the selection of data points closest to these centers.
- Extreme Y: data is sorted based on their labels, and those with the largest Y values are selected.

Table 11 presents the performance of Alg. 1 with these selection methods on Yacht and Parkinson datasets. The results indicate that the poor selection method does have a detrimental impact on our performance, particularly evident in the case of Yacht where we struggle to fit the data. In contrast, the other two sensible methods demonstrate good and comparable performance.

In order to further improve, we introduce a dynamic strategy at the end of Section 3. In this strategy, we dynamically incorporate hard samples into the support dataset. We then integrate these approaches with the proposed dynamic strategy to evaluate its effectiveness, of which the results are presented in Table 12. Based on these results, it is evident that the proposed dynamic strategy has a significantly positive impact on performance. It not only enhances accuracy but also reduces variance, resulting in more stable solutions. Even with the bad selection selection, the final performance is improved to a satisfactory level.

Table 10: The mean $R^2(\uparrow)$ and its standard derivation of Alg. 1 with respect to different batch size in SGD.

| Batch Size | 16 | 32 | 64 | 128 |
|---|---|---|---|---|
| Yacht | 0.9932±0.0065 | 0.9944±0.0044 | 0.9952±0.0047 | 0.9965±0.0019 |
| Parkinson | 0.9863±0.0041 | 0.9907±0.0027 | 0.9931±0.0018 | 0.9943±0.0018 |

Table 11: Performance of Alg.1 with different selection methods of initial support data.

| Dataset | Yacht | Yacht | Yacht | Parkinson | Parkinson | Parkinson |
|---|---|---|---|---|---|---|
| Selection Approach | Extreme Y | Y-based | X-based | Extreme Y | Y-based | X-based |
| Mean of $R^2$ | 0.0012 | 0.9957 | 0.9953 | 0.8115 | 0.9921 | 0.9928 |
| Std of $R^2$ | 0.4805 | 0.0025 | 0.0032 | 0.0126 | 0.0015 | 0.0016 |

Table 12: Performance of Alg.1 with dynamic strategy and different selection methods of initial support data.

| Dataset | Yacht | Yacht | Yacht | Parkinson | Parkinson | Parkinson |
|---|---|---|---|---|---|---|
| Selection Approach | Extreme Y | Y-based | X-based | Extreme Y | Y-based | X-based |
| Mean of R2 | 0.9961 | 0.9982 | 0.9981 | 0.9712 | 0.9972 | 0.9966 |
| Std of R2 | 0.0126 | 0.0015 | 0.0016 | 0.0049 | 0.0007 | 0.0013 |