# OpenReview forum: "Enhancing Kernel Flexibility via Learning Asymmetric Locally-Adaptive Kernels"
_ICLR.cc/2024/Conference — Submitted to ICLR 2024_

### Official Review · Reviewer_PWHQ · 2023-10-30

**Soundness:** 2 fair
**Presentation:** 3 good
**Contribution:** 3 good
**Rating:** 8
**Confidence:** 4

**Summary:**

The algorithm introduces a variant of the centered-based kernel method, they call the centers "support vectors". The idea is that the model's size remains smaller than the entire dataset, similar to  FALKON and EigenPro3.0. In contrast to methods that utilize fixed model support vectors, their algorithm adaptively adjusts these support vectors throughout the training process. Moreover, utilizing concepts from the asymmetric kernel method, they adaptively fit the support vectors with varying bandwidths throughout the training process.They show that their algorithm outperform some existing methods on several data sets.

**Strengths:**

1. the most interesting part is the idea of adaptively change the support vectors in an iterative manner. This was something novel worth exploring more.
2. The idea of adaptively adjust the bandwidth and mixing it with asymmetric kernel methods seems intriguing.(Not sure how useful)
3. The paper is clear and easy to follow.

**Weaknesses:**

Main concern:
1. The most important caveats is that the paper has only compared to vanilla KRR methods. It is not surprising that they got a slightly better performance compare for example to FALKON. I'm not at all convened this methods is better than well developed techniques such as:

i. traditional Automatic Relevance Determination(ARD) known in GP community, it is implemented with Gpytorch see here:https://docs.gpytorch.ai/en/stable/kernels.html. or see section 5 of this https://gaussianprocess.org/gpml/chapters/RW.pdf

ii. EigenPro3.0, see https://arxiv.org/abs/2302.02605

iii. Recursive Feature machines (RFMs), see: https://arxiv.org/abs/2212.13881

Scalability:

2. the authors claim that this method is scalable and they provide table 3 to justify this. But those data sets are not at all large scale. The inverse problem can be done using direct calculation for those cases. The authors should try other data sets such as Taxi, CIFAR5m to justify consistency and scalability. (both in data and model size)
3. It is mentioned in section 3 that the computation complexity is O(N_sv^3). This fundamentally shows this method on its own is not scalable. Eventually you need to scale the required support vectors(or model size) as it is discussed in https://arxiv.org/abs/2302.02605.
However, I can see that this method combined by other methods like FALKON or EigenPro3.0  can potentially be scalable.
4. How do you compute line 4 of the algorithm? Did you use FALKON or some other off the shelf algorithm or you did direct inverse?


Minor issues:

1. RBF kernel are known to be sensitive to bandwidth. While you have results for MKL, the performance of your method specifically for the Laplace kernel, which is relatively insensitive to bandwidth, remains ambiguous. Does outperforming MKL indicate superiority over merely using Laplace? The same concern applies to NTK kernels or other popular kernels.
2. I suggest more explaining for asymmetric kernels methods. For example why the inverse even exist in equation 6. or you claimed "this paper for the first time establishes an asymmetric KRR framework", but how is it different from He et. al. paper? not clear.
3. Please add what M means in the tables, helps with reading.

**Questions:**

see weaknesses

---

> ### Author Response · Authors · 2023-11-16
> **Response to Reviewer PWHQ (Part 1/2)**
>
> Thank you for your valuable suggestions and recognition of our dynamic strategy and locally adaptive bandwidths.
>
> ## Scalability is not Our Contribution
>
> First of all, we wish to emphasize our central focus, as indicated by the title. Our paper aims to explore **the application of asymmetric kernel learning in regression tasks** and assess its advantages over symmetric kernels, rather than concentrating on large datasets or large kernel machines.
>
> Our main contributions lie in (i) introducing the first asymmetric KRR model and (ii) presenting a novel asymmetric kernel learning algorithm. These two techniques collectively offer **a fresh approach to learn flexible asymmetric kernels in regression tasks**. Current investigations into asymmetric kernels continue to face **a gap in terms of interpretable regression models and efficient kernel learning algorithms**, adding to the novelty of this work.
>
> We  would like to clarify what we mean by scalability in the context of this paper. The introduction of locally-adaptive bandwidths largely enhance the flexibility of RBF kernels. This allows us to **significantly reduce the need for support data, and the decrease in support data consequently benefits the scalability of the model.** This is also what we want to emphasis in Table 3, with the reduction being more evident in the larger datasets.
>
> Consider that we can reduce the requirement for support data. When coupled with existing acceleration techniques, as you suggested, it becomes feasible for us to effectively manage millions of data points. The feasibility, however, hinges on the computing infrastructure and the number of support vectors, aspects beyond the scope of this paper.
>
> It's noteworthy that a majority of current solutions for large kernel machines are founded on symmetric kernels. We believe extending their approach to accommodate asymmetric kernels would be particularly intriguing.
>
> Nevertheless, we value your comments and will carefully revise the paper to eliminate any potential ambiguity.

---

> ### Author Response · Authors · 2023-11-16
> **Response to Reviewer PWHQ (Part 2/2)**
>
> ## Accuracy Comparison in Regression with EigenPro3.0 and RFMs
>
> In response to your concerns regarding accuracy improvement, as acknowledged by the first reviewer, our algorithms **showcase state-of-the-art accuracy in regression tasks.** Notably, our results surpass even the performance of ResNet, a benchmark for high-level regression methods in small and medium-sized datasets.
>
> We sincerely appreciate the advanced kernel methods you recommended. To address your main concern, we have conducted an additional comparative experiment with EigenPro 3.0 and Recursive Feature Machines (RFMs).
>
> **Table1: Performance of EigenPro3.0 and LAB RBF kernels on real datasets.**
> |               Dataset               |   Airfoil (N=1503)  |  Parkinson (N=5875) | Electrical (N=10000)  | KC House (N=21623) | TomsHardware (N=28179) |
> |:-----------------------------------:|:----------:|:----------:|:----------:|:--------:|:------------:|
> | Eigenpro3.0-Laplace  ($R^2$) | 0.9201 | 0.9251 | 0.9593          | 0.8636 |    0.9436    |
> |      Eigenpro3.0-#Centers       |     1202    |     4700    |     10000    |    20000   |      20000     |
> | RFMs  ($R^2$) | 0.9394 | 0.9988 |  0.9582       | 0.9008 |   0.9115    |
> |         LAB RBF kernels ($R^2$)        |   0.9608   |   0.9950   |   0.9642   |  0.8917  |    0.9809    |
> |         LAB RBF kernels (#S.V.)        |     200    |     110    |     300    |    400   |      500     |
>
>
> In our experimental setup, we train EigenPro3.0 for 50 iterations, while RFM for 5 iterations, aligning with the parameter setting outlined in their papers. As shown in the public code, RFMs use all training data points to compute the final decision function.  That is, they **use all training data as the support data.**
>
> Both EigenPro3.0 and RFM exhibit excellent capabilities in managing large-size kernel models; however, their dependence on  symmetric kernel formulations poses a high requirement on large number of support data. While by utilizing locally adaptive bandwidth, **our algorithm can significantly reduce the requirement of support data—essentially, the model complexity—while maintaining a high level of accuracy.**
>
> Moreover, we've observed that in RFMs, they learn a matrix $M$ in a generalization of the Laplace kernel $\mathcal{K}(x_i, x_j) = \exp(-\gamma\||x_i - x_j\||_M)$, where $\||x_i - x_j\||_M = (x_i - x_j)^\top M(x_i - x_j)$. That is, the matrix $M$ is designed to **capture relationships among features**.
>
> In contrast, our proposed LAB RBF kernels $\mathcal{K}(x_i, x_j) = \exp(-\||\Theta_j\odot(x_i - x_j)\||_2^2)$  and corresponding learning algorithm aim at **learning locally-adaptive bandwidths for each support data**. Exploring a completely different mechanism, combining these two methods in the future could prove to be intriguing
>
> Again, we appreciate your recommendation of these advanced methods,  and we are committed to incorporating this comparative experiment into the revised version of our work.
>
> ## Minor issues
> 1. The computation at line 4 of the algorithm, specifically evaluating $$f_{\mathcal{Z},\Theta}(t) = K_{\Theta} (t,X_{sv})(K_\Theta(X_{sv},X_{sv})+\lambda I_N)^{-1}Y_{sv}$$
> is not  time-consuming, especially when the number of support data is limited. In our experiments, we directly compute this inverse due to the manageable size of the support data. We agree that existing scalability techniques, while primarily designed for symmetric kernels, can be applied here to further speed-up the computation, opening new possibilities.
>
> 2. Distinguish from Prior Works: While previous works [[1](https://www.sciencedirect.com/science/article/pii/S1063520315001360)][[2](https://ieeexplore.ieee.org/document/10070836)] and our study use the same definition of asymmetric kernels, $\mathcal{K}(x,t) = \langle \phi(x), \psi(t)\rangle$, our paper introduces **the first formulation of asymmetric kernel ridge regression**.  In contrast to previous studies focusing on PCA and least square SVM, this results in distinct stationary solutions and applications.
> Furthermore, prior works employ manually designed asymmetric kernels such as the Kullback-Leibler Kernel and SNE Kernels (refer to [2] for details). While these kernels exhibit good performance on asymmetric data (e.g., directed graphs), their advantage on general data is not apparent. To address this limitation, our paper proposes a novel asymmetric kernel learning algorithm, demonstrating **superior performance on general data**.
>
> 3. In Eq. (6), since $(K(X,X)+\lambda I)$ is a square matrix, its inverse exists as long as it is full-rank.
> ---
> We trust that our explanation, coupled with the additional experiment, sufficiently addresses your concerns regarding both accuracy and scalability in our experiments, and we would greatly appreciate it if you could consider raising the score accordingly. We are very willing to further discuss with you on equipping current techniques to our LAB RBF kernels or any other topics that might interest you.

---

> > ### Comment · Reviewer_PWHQ · 2023-11-19
> >
> > The performance metrics you've presented are intriguing and suggest that your algorithm is competitive with other state-of-the-art (SOTA) techniques. The potential synergy of combining these methods to enhance performance is particularly noteworthy.
> >
> > This new information has significantly bolstered my confidence in your approach; previously, I had reservations about the practical utility of your algorithm. However, please make sure that,
> >
> > 1. These new results are comprehensively included in the final version of your paper.
> >
> > 2. While ARD was not tested, given its typically lesser performance compared to RFM, your results likely remain superior. Nonetheless, mentioning ARD could add value to the discussion.
> >
> > 3. It is crucial to explicitly clarify in Table 3 that your contribution is not about scalability issues.
> >
> > I trust these aspects will be addressed in the final version. Considering these new results, I believe your work is of significant value to the ICLR community. I am inclined to increase my score for your submission accordingly.

---

> ### Author Response · Authors · 2023-11-20
> **Notification of revision.**
>
> Dear reviewer PWHQ,
>
> We sincerely appreciate your understanding and recognition of our work. We are delighted that these results have boosted your confidence in our approach.
>
> The latest version of our paper is now accessible in the OpenReview system, and we hope that this updated version meets your standards. For your convenience, here we've outlined the key modifications:
>
> -   We have removed our ambiguous statement regarding scalability. Instead, in the revised version, we emphasize that learning flexible kernels can reduce the need for support data. This reduction in support data, in turn, significantly reduces model complexity, making it more efficient to deal with large-scale datasets.
> Please refer to the contributions in the end of introduction on page 4, the statements at the beginning of Section 4 on page 7, and the discussion about experimental results on larger datasets in Table 3 on page 8.
> -   EigenPro3.0 and RFMs have been introduced as new comparative methods. Please refer to Table 1 on page 7, and Table 3 on page 8 for details.
> -   A discussion on incorporating current accelerating methods for kernel machines into our methods has been added. Please refer to the line 15 on page 6.
> -   The definitions of N and M have been incorporated. Please refer to Table 1 on page 7.
> -   Additional discussions on current studies of asymmetric kernels in related works. Besides, ARD and RFMs are also included in relted works. Please refer to Section 5 on page 9.
>
> Once again, thank you for your recognition and insights. In the following days, we will continue to revise our paper according to your suggestions and those provided by other reviewers.

---

> > ### Comment · Reviewer_PWHQ · 2023-11-20
> >
> > The updated version of your paper seems sophisticated. However, I would like to point out a potential inaccuracy on page 8, under the section "Performance on large-scale datasets." The statement, "... they utilize almost all training data as support data," does not accurately represent the methodologies of FALKON and EigenPro3.0. These methods are specifically designed to avoid using the entire dataset as centers. Your approach, on the other hand, introduces an innovative iterative method for selecting centers, which effectively reduces the number of required support-vectors/centers. This distinction is crucial and should be clarified in your paper.
> >
> > Aside from this, I found your paper to be quite interesting, especially in the context of efficiently choosing centers. Consequently, I have increased my score for your paper.

---

> > > ### Author Response · Authors · 2023-11-20
> > >
> > > We sincerely appreciate your interest and recognition of our work, as well as your correction of our imprecise expression.
> > >
> > > Our intention was to convey that, in the experiments, both methods require the use of a significant portion of the training data as centers to achieve high-precision results.
> > >
> > > We are continually revising the paper based on suggestions from all reviewers, and in the final version, we will rectify this paragraph to make it clearer and more precise.
> > >
> > > Once again, we appreciate your time and continued engagement in our discussions.

---

### Official Review · Reviewer_aToc · 2023-11-01

**Soundness:** 3 good
**Presentation:** 3 good
**Contribution:** 3 good
**Rating:** 5
**Confidence:** 4

**Summary:**

This paper introduces a new asymmetric kernel names Local-Adaptive-Bandwidth RBF kernel. To solve the asymmetry of the kernel, the paper establishes an asymmetric KRR framework. To learn the kernel parameter efficiently and accelerate computation. the paper devises a kernel learning algorithm. Experimental results show the algorithm’s superiority.

**Strengths:**

1. The paper demonstrates a clear logical structure with a comprehensive framework. It tackles the complex relationship between bandwidth and data from the perspective of experimental results.
2. The paper takes into account the impact of differences in implicit mappings on the results and proposes an interesting approach to non-symmetric kernel KRR framework.
3. The paper introduces an algorithm based on dynamic strategies for parameter computation, which can effectively reduce the computational complexity associated with high-dimensional kernel matrices.

**Weaknesses:**

1. Intuitively, the relation between the mapping function's distinctiveness and the loss function, which means the coefficient of the last term in the KRR optimization objective may vary with datasets.
2. The initial data selection for support data in the kernel learning algorithm proposed in the article seems to be too random. Moreover, inappropriate data selection appears to have a significant impact on the model.

**Questions:**

1. Is the final coefficient in the asymmetric KRR framework proposed in the article required to be 1/2? Can this be understood as simply for the convenience of computing stationary points?
2. Is the small number of support vectors in the experimental results of the proposed method due to the algorithm's termination condition?

---

> ### Author Response · Authors · 2023-11-16
> **Response to Reviewer aToc (Part 1/2)**
>
> Thank you for your attentive reading and valuable suggestions.
>
> ## Coefficient of the Error Terms in Asymmetric KRR Model
> Thanks for your recognization on the novelty of our asymmetric kernel ridge regression model. We are glad that you are interested in this model. It is one of our major technical contributions to deal with asymmetric kernels on regression tasks and also provids theoretical support for subsequent asymmetric kernel learning algorithm.
>
> To address you concern on the coefficient of the error terms, it would be better to start at the LS-SVM form of our model, i.e. the model in Eq. (7).
>
> The objective function in Eq. (7) primarily incorporates a **regularization term** $w^\top v$ and a **error term** $\sum_{i=1}^N e_i r_i$. As most of machine learning models, these two terms are adjusted by **a trade-off hyper-parameter $\lambda$ varing with different datasets**.
>
> By substituting constraints in Eq. (7) to the objective function, we obtian the model in Eq. (4),  whose corresponding objective function is represented as $$\lambda w^\top v + \frac{1}{2}(\phi^\top(X)w-Y)^\top(\psi^\top(X)v-Y). $$To enhance clarity, we delved deeper into the interpretation of $(\phi^\top(X)w-Y)^\top(\psi^\top(X)v-Y)$, outlining it as two loss terms and one mapping function’s distinctiveness term. That is,
> $$2(\phi^\top(X)w-Y)^\top(\psi^\top(X)v-Y) = \||\psi^\top(X)v-Y\||_2^2 + \||\phi^\top(X)w-Y\||_2^2 - \||\psi^\top(X) v - \phi^\top(X) w\||_2^2.$$
> Therefore, **the three terms** $\||\psi^\top(X) v - \phi^\top(X) w\||_2^2$, $\||\psi^\top(X)v-Y\||_2^2$, and $\||\phi^\top(X)w-Y\||_2^2$ **are not independent; instead, they are integrated as a whole** and thus do not require a extra hyper-parameter tuning.
>
> Thank you for this insightful question. We will make it clearer in the revised version.
>
>
> ## Impact of the Initial Support Data Selection
> Firstly, you are correct in pointing out the significant influence of support data selection on the algorithm. In light of this, **we have introduced a dynamic strategy aimed at mitigating the impact of initial support data selection.**
>
> To address your concern, we explore three different approaches to initial data selection: two rational methods (Y-based and X-based) and one irrational method (Extreme Y).
>  -   Y-based (utilized in the manuscript): data is sorted based on their labels, and support data is uniformly selected.
> -   X-based: k-means is applied to the training data to identify cluster centers, followed by the selection of data points closest to these centers.
> -   Extreme Y: data is sorted based on their labels, and those with the largest Y values are selected.
>
> **Table1: Performance of Alg.1 with different selection of initial support data.**
> |Dataset| Selection Approach |  Mean of $R^2$   |   Std of $R^2$   |
> |:------------------:|:----------------:|:-------:|:------:|
> Yacht|     Extreme Y    | 0.0012 | 0.4805 |
> Yacht|     Y-based    | 0.9957 | 0.0025 |
> Yacht|    X-based    | 0.9953 | 0.0032 |
> Parkinson|     Extreme Y    |   0.8115 | 0.0126 |
> Parkinson|     Y-based    |   0.9921 | 0.0015 |
> Parkinson|    X-based    |   0.9928  |          0.0016|
>
> The results indicate that the poor selection method does have a detrimental impact on our performance, particularly evident in the case of Yacht where we struggle to fit the data. In contrast, the other two sensible methods demonstrate good and comparable performance.
>
> In order to further improve, we introduce a dynamic strategy at the end of Section 3. In this strategy, we dynamically incorporate hard samples into the support dataset. We then integrate these approaches with the proposed dynamic strategy to evaluate its effectiveness.
>
> **Table 2: Performance of Alg.1 with dynamic strategy**
> |Dataset| Selection Approach | Mean of $R^2$   |   Std of $R^2$   |
> |:------------------:|:----------------:|:-------:|:-------:|
> Yacht|   Extreme Y     | 0.9961 | 0.0126 |
> Yacht|     Y-based    | 0.9982 | 0.0015 |
> Yacht|    X-based    |  0.9981 | 0.0016 |
> Parkinson|   Extreme Y     | 0.9712 | 0.0049  |
> Parkinson|     Y-based    | 0.9972 | 0.0007  |
> Parkinson|    X-based    |  0.9966 | 0.0013  |
>
> Based on these results, it is evident that the proposed dynamic strategy has a significantly positive impact on performance. It not only enhances accuracy but also reduces variance, resulting in more stable solutions. Even with the bad selection selection, the final performance is improved to a satisfactory level.

---

> ### Author Response · Authors · 2023-11-16
> **Response to Reviewer aToc (Part 2/2)**
>
> ## Why is the Number of Support Data is so Small?
>
> In fact, the limitation in support data is not attributed to the termination condition. Instead, the constrained support data stems from **the remarkable flexibility of LAB RBF kernels.** This flexibility empowers our algorithm to represent intricate data patterns with a reduced number of support data.
>
> In determining the termination condition, we prefer achieving a modest number of support data, given its advantages in computational efficiency and memory conservation, all while upholding accuracy.
>
> Traditional symmetric kernels, lacking such flexibility, struggle to achieve the same efficiency as they apply a uniform kernel formula to all data points.
>
> The example in Figure 1 in the manuscript precisely illustrates this point. Attempting to fit unevenly distributed data with a uniform bandwidth necessitates more support data to approximate local distributions, as depicted in (b) and (c). By leveraging the high flexibility of LAB RBF kernels, as demonstrated in (d), our algorithm perfectly tackles this challenge, resulting in advantages in both accuracy and model complexity.
>
> ---
> We hope that our explanations and the additional experimental results adequately address your concerns regarding support data selection and the formulation of the asymmetric KRR model. We would greatly appreciate it if you could consider revising the score accordingly. Moreover, we are open to further discussions regarding the design of the asymmetric KRR model or any other topics that might interest you.

---

> ### Author Response · Authors · 2023-11-20
> **Notification of revision.**
>
> Dear reviewer aToc,
>
> Just now, we have uploaded the revised papar in OpenReview system and sincerely hope it meets your standards. Here is a quick look of the key updates:
>
>  - We tried our best to better clarify the design of the asymmetric kernel ridge regression model in Eq.(4). Please review the reorgnized sentences in Section 2.2 on page 4.
> -   We added the discussion and the additional experiment on the selection of initial support data in Appendix F. We trust these additions will provide robust evidence for the effectiveness of our proposed dynamic strategy.
>
> We're truly grateful for your valuable suggestions, which have played a pivotal role in elevating the quality of our paper.  Any lingering concerns on your end? We're all ears and more than ready to tackle them before the discussion period wraps up.

---

> > ### Comment · Reviewer_aToc · 2023-11-23
> >
> > I truely appreciate your detailed response and additional experiments, and they are helpful to me to better understand your work. However, I still think there are some major concerns which may not be solved in a short time:
> > 1) The application prospect of this research is not clear, in what scenario the proposed asymmetric kernel can produce better performance?
> > 2) It seems a trivial study on kernel methods area, the paper only proposed an asymmetric kernel ridge regression model and a gradient based optimization method, the computation cost is still high. I think the research may not provide some helpful insights on kernel method area.
> > 3) The kernel ridge regression is a classical model, but not a good choice in most applications at present, while the deep models are widely applied in many areas. However, the proposed asymmetric kernel seems can not incorporate with deep models, which makes the study shows less significance.

---

> > > ### Author Response · Authors · 2023-11-23
> > >
> > > Thank you for investing your time in our discussion and for sharing valuable insights on asymmetric kernels. Regrettably, due to time constraints, we would like to briefly address your concerns.
> > >
> > > ## In What Scenario Asymmetric Kernels can Outperform?
> > >
> > > We designed our algorithm to be applicable in various scenarios, evident in our choice of experiment settings and datasets that were not specially tailored. Their results clearly demonstrate that learning flexible asymmetric kernels significantly outperforms symmetric kernels, excelling in both accuracy and model complexity.
> > >
> > > ## Contributions to Kernel Methods Area
> > > In the realm of kernel methods, the quest for flexible or learnable kernels has been an ongoing exploration, encompassing areas like multiple kernel learning and deep kernel learning. Notably, **our methods explore a completely different approach to enhance kernel flexibility by focusing on learning asymmetric kernels.**
> > >
> > > The success of our asymmetric kernel learning algorithm not only presents an alternative strategy but also answers a longstanding question: **Are models in the reproducible kernel Banach space (RKBS) superior to those in the reproducible kernel Hilbert space (RKHS)?** While theoretical studies on RKBS for asymmetric kernels have been ongoing, a practical and powerful algorithm has been lacking. Our study bridges this gap, and we anticipate it will garner positive feedback for theoretical research on asymmetric kernels.
> > >
> > > ## Link to Deep Models
> > >  **It is precisely because we have achieved training asymmetric kernels using gradient descent methods that the combination of asymmetric kernels and deep learning has become feasible.** In our forthcoming research, we plan to fuse our learnable asymmetric kernel with deep architectures. Trained through gradient descent, our learnable kernel can be seen as a specialized block that can seamlessly integrate into other architectures. Whether it involves adding feature learning layers before (similar to some existing deep kernel learning methods) or incorporating downstream tasks afterward, our learnable kernels are versatile enough to handle both scenarios.
> > >
> > > In short, we are actively exploring potential synergies between kernel methods and deep neural networks. The gradient-descent-based kernel learning framework presented here serves as a significant first step in this exploration.
> > >
> > >  ----
> > > In addition to the aforementioned discussion, we have numerous interesting and meaningful plans for further exploring asymmetric kernel learning. We hope to convey our thoughts and research findings on asymmetric kernels to the community through this paper. Once again, we sincerely appreciate your time and ongoing engagement in discussions with us.

---

> > > ### Author Response · Authors · 2023-11-23
> > >
> > > We have recently submitted a revised version of the manuscript. In the conclusion section, we have included additional discussions on asymmetric kernels and Transformers, as well as insights into the future potential of our approach. We appreciate your feedback, which has greatly contributed to enhancing the overall quality of the paper.

---

### Official Review · Reviewer_gJwQ · 2023-11-03

**Soundness:** 3 good
**Presentation:** 3 good
**Contribution:** 3 good
**Rating:** 8
**Confidence:** 4

**Summary:**

The paper proposes a novel approach to enhance the flexibility of kernel-based learning by introducing Locally-Adaptive-Bandwidth (LAB) kernels. Unlike traditional fixed kernels, LAB kernels incorporate data-dependent bandwidths, allowing for better adaptation to diverse data patterns. To address challenges related to asymmetry and learning efficiency, the paper introduces an asymmetric kernel ridge regression framework and an iterative kernel learning algorithm. Experimental results demonstrate the superior performance of the proposed algorithm compared to existing methods in handling large-scale datasets and achieving higher regression accuracy.

**Strengths:**

1. The introduction of LAB kernels with trainable bandwidths significantly improves the flexibility of kernel-based learning. By adapting bandwidths to individual data points, the model can better accommodate diverse data patterns, leading to more accurate representations.
2. The paper establishes an asymmetric kernel ridge regression framework specifically designed for LAB kernels. Despite the asymmetry of the kernel matrix, the stationary points are elegantly represented as a linear combination of function evaluations at training data, enabling efficient learning and inference.
3. The proposed algorithm allows for the estimation of bandwidths from the training data, reducing the demand for extensive support data. This data-driven approach enhances generalization ability by effectively tuning bandwidths based on the available training data.
4. The proposed algorithm shows superior scalability in handling large-scale datasets compared to Nyström approximation-based algorithms. LAB kernels, with their adaptive bandwidths, offer a flexible and efficient solution for kernel-based learning tasks with extensive data.

**Weaknesses:**

1. While the paper presents empirical evidence of the superior performance of the proposed algorithm, it may lack strong theoretical guarantees or formal analysis of its convergence properties. Further theoretical investigations may be needed to fully understand the behavior and limitations of LAB kernels
2. The performance of LAB kernels heavily relies on the accurate estimation of bandwidths. Selecting appropriate bandwidths for different data patterns can be a challenging task, and suboptimal choices may result in reduced performance or overfitting.

**Questions:**

See Weaknesses.

---

> ### Author Response · Authors · 2023-11-16
> **Response to Reviewer gJwQ (Part 1/2)**
>
> We sincerely appreciate your valuable comments and highly recognition of our work.
>
> ## About Convergence Properties
> We agree with you regarding the significance of convergence properties in algorithmic discussions. Given that our optimization problem, as outlined in equation (9), is a non-convex unconstrained challenge, we have opted for the utilization of stochastic gradient descent (SGD) methods to efficiently address it. Consequently, the convergence analysis of SGD in the realm of general non-convex optimization is pertinent to our algorithm, e.g.[[1]](https://dl.acm.org/doi/abs/10.5555/3455716.3455852).
> For this reason, we chose not to delve into this topic in the manuscript. But we value your suggestions and intend to incorporate a discussion on it in the revised version.
>
> [1] Fehrman, Benjamin, Benjamin Gess, and Arnulf Jentzen. "Convergence rates for the stochastic gradient descent method for non-convex objective functions." _The Journal of Machine Learning Research_ 21, no. 1 (2020): 5354-5401.
>
> ## Learning Behavior of LAB RBF Kenrels
>
> The learning behavior of LAB RBF kernels, as you point out, is indeed both intriguing and notably distinct from existing kernels due to their inherently asymmetric nature. The loss of symmetry renders the analysis tools in traditional RKHS, or even in more general case e.g., reproducing kernel Kre&iuml;n spaces (RKKS) and reproducing kernel Banach spaces (RKBS), not applicable. Our ongoing work specifically addresses and explores these unique characteristics.
>
> Notably, LAB RBF kernels, featuring trainable bandwidths, share a resemblance with multiple kernel learning whose corresponding functional space comprises a direct sum space of Hilbert spaces. Inspired by this correlation, our analysis of LAB RBF kernels aims to construct a sparse model within the direct integral space of Hilbert spaces. This work is not only interesting but also presents a notable challenge due to the complex nature of the space being explored.

---

> ### Author Response · Authors · 2023-11-16
> **Response to Reviewer gJwQ (Part 2/2)**
>
> ## Effects of Inaccurate Estimation of Bandwidths
>
> We appreciate your insights into the crucial role that bandwidths play in the performance of LAB RBF kernels. It is precisely for this purpose, we propose the kernel learning algorithm to determine proper bandwidths for different data patterns.
>
> Indeed there might be some challenges to determine good bandwidth in practice. For example, within our kernel learning algorithm, various factors during training, such as the choice of initial points, stopping criteria, learning rate, and batch size, may result in distinct bandwidth estimations.
>
> In our experiments, these hyper-parameters are determined through cross-validation, detailed in Appendix C. And the influence of initialization is thoroughly addressed in Appendix E.
>
> Here, to address you concern, we present extra experimental results that demonstrate their impact on the performance of our algorithm.
>
> **Impact of Maximal Iteration Number**
> 1. Yacht dataset: 246 training data, batch size=32.
>
> | Iterations | 0      | 100    | 200    | 300    | 400    | 500    | 1000   | 2000   | 3000   | 4000   | 5000   |
> |:--------:|:--------:|:--------:|:--------:|:--------:|:--------:|:--------:|:--------:|:--------:|:--------:|:--------:|:--------:|
> | Mean of $R^2$ | 0.5633 | 0.9396 | 0.9789 | 0.9872 | 0.9902 | 0.9913 | 0.9940  | 0.9956 | 0.9966 | 0.9966 | 0.997  |
> | Std of $R^2$  | 0.0737 | 0.0316 | 0.0187 | 0.0105 | 0.0073 | 0.0065 | 0.0056 | 0.0033 | 0.0023 | 0.0024 | 0.0021 |
>
> 2. Pakinson dataset: 4700 training data, batch size=128.
>
> | Iterations| 0      | 100    | 200    | 300    | 400    | 500    | 1000   | 2000   | 3000   | 4000   | 5000   |
> |:--------:|:--------:|:--------:|:--------:|:--------:|:--------:|:--------:|:--------:|:--------:|:--------:|:--------:|:--------:|
> | Mean of $R^2$ | 0.5458 | 0.9101 | 0.9445 | 0.9622 | 0.9752 | 0.9814 | 0.9889 | 0.9895 | 0.9925 | 0.9931 | 0.9944 |
> | Std of $R^2$  | 0.0315 | 0.0065 | 0.0061 | 0.0052 | 0.0061 | 0.0055 | 0.0033 | 0.0023 | 0.0026 | 0.0019 | 0.0019 |
>
> **Impact of Learning Rate**
> 1. Yacht dataset
>
> |Learning Rate| 1.00E+00 | 1.00E-01 | 1.00E-02 | 1.00E-03 | 5.00E-04 | 1.00E-04 |
> |:--------:|:--------:|:--------:|:--------:|:--------:|:--------:|:--------:|
> | Mean of $R^2$ |  0.9723 |  0.9968 |  0.9970 |  0.9911 |  0.9804 |  0.8089  |
> |Std of $R^2$|  0.0164 |  0.0019 |  0.0017 |  0.0038 |  0.0104 |  0.0246  |
>
> 2. Pakinson dataset:
>
> | Learning Rate | 5.00E-01 | 1.00E-01 | 5.00E-02 | 1.00E-02 | 5.00E-03 | 1.00E-03 |
> |:--------:|:--------:|:--------:|:--------:|:--------:|:--------:|:--------:|
> |  Mean of $R^2$  | 0.98275  | 0.99411  | 0.9943   | 0.99066  | 0.9763   | 0.8947   |
> | Std of $R^2$  | 0.0031   | 0.0018   | 0.0018   | 0.0022   | 0.0058   | 0.0052   |
>
> **Impact of Batch Size**
> 1. Yacht dataset
>
> |  Batch Size | 16       | 32       | 64        | 128      |
> |:----------:|----------|----------|-----------|----------|
> | Mean of $R^2$ | 0.9932 | 0.9944 | 0.9952 | 0.9965 |
> |  Std of $R^2$ | 0.0065 | 0.0044 | 0.0047 | 0.0019 |
>
> 2. Pakinson Dataset:
>
> | Batch Size | 16      | 32      | 64      | 128    |
> |:----------:|----------|----------|-----------|----------|
> | Mean of $R^2$  | 0.9863 | 0.9907 | 0.9931 | 0.9943 |
> |  Std of $R^2$  | 0.0041 | 0.0027 | 0.0018 | 0.0018 |
>
> These results underscore that the carefully selection of these hyperparameters enhances the final performance. Nevertheless, even under suboptimal hyperparameter settings, the performance remains commendable, albeit with varying bandwidth estimates. This highlights the robustness and insensitivity of our algorithm across a wide spectrum of hyperparameter choices.
>
>
> ---
> We hope that our explanations, along with the additional experiments, effectively address your concerns regarding algorithm convergence and robustness. Furthermore, we welcome the opportunity for further discussions on the learning behavior of LAB RBF kernels if you are interested.

---

> ### Author Response · Authors · 2023-11-20
> **Notification of revision.**
>
> Dear reviewer gJwQ,
> We've just uploaded the revised paper onto the OpenReview system. Taking your insightful suggestions into account, we've made thorough revisions to the manuscript, where you can find that:
>
>  - On page 7, we added the discussion of convergence analysis in the end of Section 3.
>  - In the Appendix E, we added the extra experiments on the effects of hyper-parameters on final performance.
>
> Once again, we appreciate these valuable suggestions from you, which have significantly contributed to enhancing the quality of our paper.

---

### Author Response · Authors · 2023-11-16
**Common Response**

Dear Program Chairs, Area Chairs, and Reviewers,

First and foremost, we extend our sincere gratitude to the reviewers for dedicating their time to thoroughly evaluate our work and for providing constructive critiques and valuable suggestions. We highly appreciate the insightful comments that have been instrumental in improving the quality of our manuscript. In response to the reviewers' comments, we have diligently worked towards addressing each point raised. The detailed responses to individual comments are outlined below.

**We look forward to further refining our paper based on the continued feedback, so the updated version will be uploaded in two days.** All additional experiments conducted in response to the reviewers' feedback will be added in the manuscript or the appendix. Furthermore, we will make the corresponding code publicly accessible after the completion of the review process. We believe that these supplementary materials will contribute to the transparency and reproducibility of our research.

---

### Author Response · Authors · 2023-11-22

Dear Reviewers,

Thank you for your valuable feedback. We have released a new version of our paper, primarily addressing the removal of highlighted portions in accordance with the ICLR guidelines. Additionally, we have refined imprecise expressions.

As the discussion period continues, we look forward to hearing your valuable insights. Your constructive suggestions are greatly appreciated, and we are committed to making continuous revisions to ensure our paper reaches its best possible final version.

---

### Comment · Area_Chair_8xnv · 2023-11-22
**Let's have (more) discussion with authors**

Dear reviewers,

The author-reviewer discussion period is closing at the end of Wednesday Nov 22nd (AOE). Let's take this remaining time to have (more) discussions with the authors on their responses to your reviews. Should you have any further opinions, comments or questions, please let the authors know asap and this will allow the authors to address them.

Kind regards, AC

---

### Meta-Review · Area_Chair_8xnv · 2023-12-09

**Metareview:**

Based on the submission, reviews, author feedback, and discussions, the main points that have been raised are summarised as follows.

Strengths:

1. A kernel with data-dependent bandwidth is proposed.
2. An asymmetric kernel ridge regression framework is developed.
3. The idea of adaptively selecting the support vectors/centers is interesting and innovative.
4. The paper is easy to follow and has a clear logical structure.

Issues:

1. Need theoretical investigation to fully understand the proposed kernel.
2. The impact of data selection and the cause of the small number of support vectors need to be further clarified.
3. The scenario in which the proposed asymmetric kernel can produce better performance is not clear.
4. New experimental results need to be comprehensively included.

In addition, AC raises the following issues for discussion.

1. The idea of asymmetric kernel has been studied in the literature. In a work [R1] (cited in this submission), an asymmetric kernel-based learning framework is proposed for the least squares support vector machine (LS-SVM). Additionally, it is further found that this submission misses existing related work that exactly applies asymmetric kernel to regression task [R2, R3]. The relationship and advantage w.r.t to existing work is not clear.

2. The dynamic strategy has a significant impact to the performance of the proposed method. However, the role of this dynamic strategy is not specifically clarified by experimental study.  Also, the dynamic strategy is not fundamentally new.

3. The experimental study does not seem to be convincing enough. The difference among the methods in comparison sometime is too small (e.g., In Table 1, the difference is small for the datasets of Yacht, SML, and Comp-active.). Also, the proposed work is not compared with the existing ones (e.g., in related work) that also focus on asymmetric kernel.

AC further discussed this work with SAC and then made the final recommendation.

[R1] Learning with Asymmetric Kernels: Least Squares and Feature Interpretation, He et al., IEEE TPAMI 2023

[R2] Asymmetric kernel in Gaussian Processes for learning target variance, Pintea et al., Pattern Recognition Letters, 2018

[R3] Asymmetric kernel regression, M. Mackenzie and A. K. Tieu, IEEE TNN, 2004

**Justification For Why Not Higher Score:**

1. The most critical issue: The related work in the literature is not sufficiently considered in this work. The novelty and contribution is significantly impacted by taking into account the related work.
2. The cause of the small number of support vectors is not clarified. It is not clear whether the merit of having a smaller number of support data is really attributed to the proposed kernel or due to the use of this dynamic strategy.
3. The proposed dynamic strategy is not fundamentally new. Conceptually, it is closely related to hard sample mining and has been practically used in the literature of neural network training or AdaBoost training to emphasis difficult training data.

**Justification For Why Not Lower Score:**

N/A

---

### Decision · Program_Chairs · 2024-01-16

Reject